# Transient intracellular acidification regulates the core transcriptional heat shock response

Catherine G Triandafillou[1], Christopher D Katanski[2], Aaron R Dinner[3], D Allan Drummond[2]*

[1]Graduate Program in Biophysical Sciences, The University of Chicago, Chicago, United States; [2]Department of Biochemistry and Molecular Biology and Department of Medicine, Section of Genetic Medicine, The University of Chicago, Chicago, United States; [3]Department of Chemistry and the James Franck Institute, The University of Chicago, Chicago, United States

**Abstract** Heat shock induces a conserved transcriptional program regulated by heat shock factor 1 (Hsf1) in eukaryotic cells. Activation of this heat shock response is triggered by heat-induced misfolding of newly synthesized polypeptides, and so has been thought to depend on ongoing protein synthesis. Here, using the budding yeast *Saccharomyces cerevisiae*, we report the discovery that Hsf1 can be robustly activated when protein synthesis is inhibited, so long as cells undergo cytosolic acidification. Heat shock has long been known to cause transient intracellular acidification which, for reasons which have remained unclear, is associated with increased stress resistance in eukaryotes. We demonstrate that acidification is required for heat shock response induction in translationally inhibited cells, and specifically affects Hsf1 activation. Physiological heat-triggered acidification also increases population fitness and promotes cell cycle reentry following heat shock. Our results uncover a previously unknown adaptive dimension of the well-studied eukaryotic heat shock response.

*For correspondence:
dadrummond@uchicago.edu

Competing interests: The authors declare that no competing interests exist.

## Introduction

To survive and thrive, organisms must rapidly respond when their environments turn harsh. Cells across the tree of life possess the capacity to adaptively respond to primordial stresses—heat, starvation, hypoxia, noxious compounds—in a conserved program involving the production of so-called heat shock proteins, many of which act as molecular chaperones (*Lindquist, 1986*). Transcription of heat shock genes surges at the onset of stress, reaching as much as a thousand fold during thermal stress, with more modest induction accompanying nutrient withdrawal and diverse other stresses (*Lindquist, 1986*; *Zid and O'Shea, 2014*; *Morano et al., 2012*; *Gidalevitz et al., 2011*). In eukaryotes, the transcriptional stress response is controlled by multiple factors, with the heat shock transcription factor Hsf1 regulating induction of a core group of chaperones (*Solís et al., 2016*; *Pincus et al., 2018*). Basal levels of chaperones repress Hsf1 by direct binding (*Shi et al., 1998*; *Zheng et al., 2016*; *Krakowiak et al., 2018*), and removal of this repression in the absence of stress suffices to activate transcription (*Zheng et al., 2016*; *Krakowiak et al., 2018*). Induced chaperones, in turn, assist with protein folding, as well as preventing and dispersing stress-induced molecular aggregates (*Vabulas et al., 2010*; *Richter et al., 2010*; *Cherkasov et al., 2013*; *Walters et al., 2015*; *Kroschwald et al., 2015*; *Kroschwald et al., 2018*).

The mechanism by which the Hsf1-mediated transcriptional response is induced following physiological heat shock is incomplete and has remained so since the response's discovery nearly 60 years ago (*Ritossa, 1962*). In the currently accepted model for heat-triggered Hsf1 activation, events

proceed as follows. Hsf1 is constitutively bound and repressed by the molecular chaperone Hsp70 before stress (**Krakowiak et al., 2018**; **Zheng et al., 2016**). Heat stress is thought to cause deleterious protein unfolding (**Richter et al., 2010**) (misfolding) which exposes hydrophobic regions (**Vabulas et al., 2010**) for which Hsp70 has high affinity (**Rüdiger et al., 1997**). Titration of Hsp70 away from Hsf1 suffices to induce Hsf1 (**Zheng et al., 2016**; **Krakowiak et al., 2018**). Despite the crucial role misfolded proteins play in this model, no specific endogenous eukaryotic protein has been demonstrated to misfold in vivo in response to a sublethal heat shock. Instead, newly synthesized polypeptides (which include nascent chains still being synthesized and complete polypeptides which have yet to reach their native structure) are thought to serve as Hsf1 inducers during heat stress (**Baler et al., 1992**; **Tanabe et al., 1997**; **Li et al., 2017**). Suppression of newly synthesized polypeptides by translation inhibition suppresses the heat-induced transcription of genes regulated by Hsf1 (**Baler et al., 1992**; **Beckmann et al., 1992**; **Albert et al., 2019**; **Masser et al., 2019**). Consequently, ongoing translation has been deemed a requirement for Hsf1 activation (**Masser et al., 2019**).

Notably, the same diverse environmental changes which stimulate the transcriptional response are also accompanied by intracellular acidification—a drop in cytosolic pH (**Weitzel et al., 1987**; **Bright and Ellis, 1992**; **Munder et al., 2016**; **Kroschwald et al., 2015**). Like the transcriptional response, stress-induced acidification is broadly conserved in eukaryotes, including mammals (**Bright and Ellis, 1992**; **Yao and Haddad, 2004**; **Tombaugh and Sapolsky, 1993**; **Díaz et al., 2016**), insects (**Drummond et al., 1986**; **Zhong et al., 1999**), plants (**Ishizawa, 2014**), and fungi (**Weitzel et al., 1987**; **Kroschwald et al., 2015**). Although acidification has sometimes been viewed as a toxic consequence of stress, particularly in studies of hypoxia and ischemia-associated acidosis (**Ishizawa, 2014**; **Tombaugh and Sapolsky, 1993**), the cytoprotective effects of short-term acidification were identified decades ago (**Tombaugh and Sapolsky, 1993**). Recent work has shown that interfering with energy-depletion-induced acidification in budding yeast and in fission yeast, which diverged from budding yeast more than half a billion years ago (**Hedges et al., 2015**), compromises the fitness of both species (**Munder et al., 2016**; **Joyner et al., 2016**). Furthermore, many mature proteins associated with stress-induced condensation show a strong dependence on pH for their self-association, whether by polymerization or phase separation (**Petrovska et al., 2014**; **Kroschwald et al., 2015**; **Munder et al., 2016**; **Riback et al., 2017**; **Franzmann et al., 2018**).

What role does stress-induced cellular acidification play in the transcriptional response to heat shock? Early work in *Drosophila melanogaster* produced mixed results: one study indicated that acidification had little impact on the production of heat shock proteins (**Drummond et al., 1986**), while later work showed that Hsf1 trimerization, a key activation step, could be induced by acidification in vitro (**Zhong et al., 1999**). More recently, acidification during stress has been shown to influence cell signaling (**Dechant et al., 2010**; **Gutierrez et al., 2017**) and appears to be cytoprotective (**Munder et al., 2016**; **Joyner et al., 2016**; **Coote et al., 1991**; **Panaretou and Piper, 1990**). The extent to which this adaptive effect of pH depends on the core transcriptional stress response remains unknown. What has been demonstrated is that cell cycle reentry after heat shock follows the dissolution of stress granules, which depends on the products of stress-induced transcriptional changes: molecular chaperones (**Kroschwald et al., 2015**). These results suggest a clear link between stress-triggered transcription of heat shock genes and growth. Exactly how do intracellular acidification, transcriptional induction, chaperone production, and cellular growth interrelate following heat shock?

To answer this question, we developed a single-cell system to both monitor and manipulate cytosolic pH while tracking the induction of molecular chaperones in budding yeast. We find that acidification universally promotes the heat shock response, and that when canonical triggers for the response—the newly synthesized polypeptides—are suppressed, acidification is required for cells to respond to heat shock. Acidification alone, however, is insufficient to induce a response. We measure fitness on both the population and single-cell level and find that in both cases, the physiological stress-associated drop in pH promotes fitness. Global measurement of transcript levels as a function of intracellular pH during heat shock reveals specific suppression of core Hsf1 target genes when intracellular acidification is prevented.

The mechanism underlying Hsf1's pH-dependent activation remains open. However, our results are consistent with a previous hypothesis positing a role for temperature- and pH-dependent phase

separation in sensing heat stress (*Riback et al., 2017*), leading us to predict a specific mechanism in which elevated pH suppresses a temperature-sensitive phase separation process.

Our results link cytosolic acidification to the regulation of the canonical transcriptional heat shock response and subsequent stress adaptation in single cells, indicating that pH regulation plays a central role in the Hsf1-mediated stress response.

## Results

### A high-throughput assay allows quantification of single-cell responses to heat shock

Yeast thrive in acidic environments, and spend significant cellular resources on the activity of membrane-associated proton pumps which keep the cytoplasm at a resting pH of around 7.5 (*Orij et al., 2011*). The resulting electrochemical gradient is used to drive transport and other crucial cellular processes, but is disrupted during stress, causing cells to acidify (*Figure 1*). While the mechanism of proton influx remains poorly understood, elevated temperature increases membrane permeability (*Coote et al., 1994*) and other stresses have been shown to reduce proton pump activity (*Orij et al., 2011*; *Orij et al., 2012*; *Dechant et al., 2010*). We first sought to precisely measure the intracellular pH changes associated with heat stress.

To track intracellular pH during stress and recovery, we engineered yeast cells to constitutively express pHluorin, a pH biosensor derived from green fluorescent protein (*Miesenböck et al., 1998*), in the cytoplasm. The probe was calibrated to known pH values in vivo (*Figure 1—figure supplement 1* and Materials and methods). We used this strain to characterize intracellular pH changes occurring during heat stress and recovery. During a 42°C, 10 min heat stress in acidic media (pH 4) we find that cells rapidly and reproducibly acidify from a resting pH of approximately 7.5 to a range of slightly acidic pH values around 6.8 to 7.0 (*Figure 1B*, *Figure 1—figure supplement 1C*, in agreement with previous results [*Weitzel et al., 1987*]). When returned to normal growth temperature (30°C), cells restore the resting pH in approximately ten minutes. The minimum pH reached is similar for cells stressed at 42°C for 20 min (*Figure 1—figure supplement 1D*).

The hallmark of the heat shock response is the production of molecular chaperones (*Lindquist, 1986*; *Vabulas et al., 2010*; *Morano et al., 2012*). To assess the effects of acidification on this response, we measured chaperone induction by engineering a pHluorin-labeled yeast strain to express a red-fluorescent-protein-tagged version of Ssa4 (Ssa4-mCherry) from the endogenous *SSA4* locus. Ssa4 is a strongly temperature-responsive Hsp70 chaperone, and its encoding gene is a specific Hsf1 target (*Hottiger et al., 1992*; *Pincus et al., 2018*; *Morano et al., 2012*). This two-color reporter strain allowed us to simultaneously track intracellular pH and the stress response at the single-cell level.

We stressed cells at 42°C for 20 min and then returned them to 30°C to recover. Samples were collected at 15- to 30 min intervals during recovery and analyzed by flow cytometry to monitor Ssa4-mCherry production. An example of the raw data, showing an increase in fluorescence in the mCherry channel over time, is shown in *Figure 1C*. Although the appearance of a fluorescent signal is delayed by the maturation time of the fluorophore, mCherry, confounds determination of the absolute timing of the response, this delay is shared across experiments, allowing for direct comparison between conditions and replicates. For each independent experiment, we tracked the median relative change in red fluorescence over time, creating induction curves which characterize the response, as in *Figure 1D*.

### Intracellular acidification during heat shock promotes rapid heat-shock protein production

With the tools in hand to quantify intracellular pH and induction of stress proteins, we set out to first determine whether acidification during stress affected the cellular response. Existing evidence (*Orij et al., 2011*) indicates that acidification results primarily from an influx of environmental protons, rather than (for example) the release of protons from internal stores such as the vacuole. We confirmed a dependence on external protons by heat-stressing cells in normal, acidic media (pH 4), or in media where the pH had been adjusted to the cellular resting pH (7.5). Stressing cells in non-

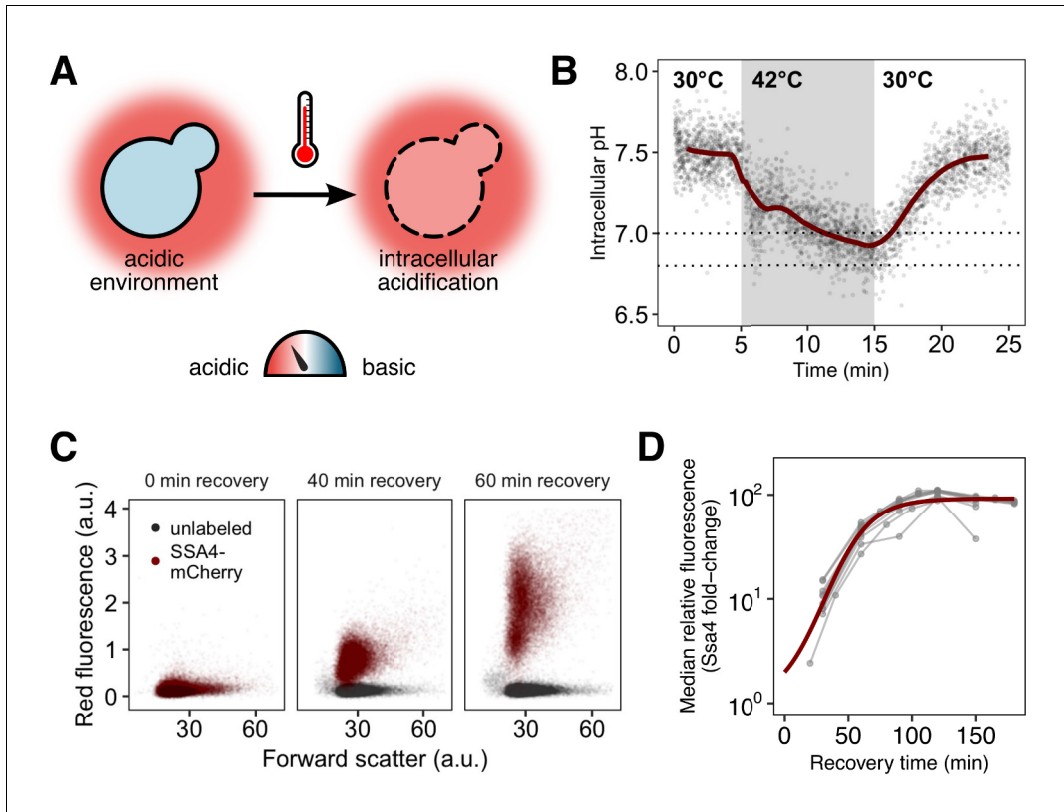

**Figure 1.** Yeast cells respond to heat shock with intracellular pH changes and production of heat-shock proteins, which can be tracked at the single-cell level. (**A**) *S. cerevisiae* cells live in acidic environments but maintain a neutral or slightly basic intracellular pH. During heat stress the cell membrane becomes more permeable, leading to intracellular acidification. (**B**) Intracellular pH changes during stress measured with continuous flow cytometry. Each point is an individual cell. The gray region is the period during which cells were exposed to elevated temperature. A solid line shows a sliding-window average over all data; for visual clarity, only 2% of points are shown. Dashed lines represent the range we subsequently use as representative of the physiological pH drop. (**C**) Induction of labeled Hsp70 (Ssa4-mCherry) after heat shock. Each plot is a timepoint during recovery from 42°C, 20 min heat shock showing forward scatter pulse area, which correlates roughly with size, versus red fluorescence. Unlabeled cells are shown in black for comparison. (**D**) Summary of induction of Ssa4-mCherry after heat shock; each point represents the fold change, relative to unstressed cells, of the median fluorescence of >5000 cells expressed as a ratio to forward scatter; each gray line is an experiment ($n = 6$). Thick red curve is a sigmoid fit (see Materials and methods).

The online version of this article includes the following figure supplement(s) for figure 1:

**Figure supplement 1.** pHluorin calibration curve and replicates of B.

acidic media prevented acidification (*Figure 2A*). Cells that could not acidify during stress delayed and reduced the induction of Ssa4 (*Figure 2D*, left hand side).

Misfolding of newly synthesized polypeptides is thought to provide the primary trigger for Hsf1 activation, as described in the Introduction. To test whether acidification still promoted the stress response even under conditions where the concentration of newly synthesized polypeptides would be sharply limited, we first used brief glucose withdrawal, a physiologically relevant condition which is known to rapidly and reversibly inhibit translation of most cellular mRNAs (*Ashe et al., 2000* and *Figure 2B*). We heat-stressed cells, then returned them to favorable growth conditions (2% glucose, 30°C) to recover. Strikingly, we found that even in the absence of translation and presumably misfolded newly synthesized polypeptides, cells that could acidify during stress responded almost identically to cells stressed while global translation was unperturbed. However, cells that were not actively translating and also were unable to acidify during stress almost completely failed to respond (*Figure 2D*, right hand side).

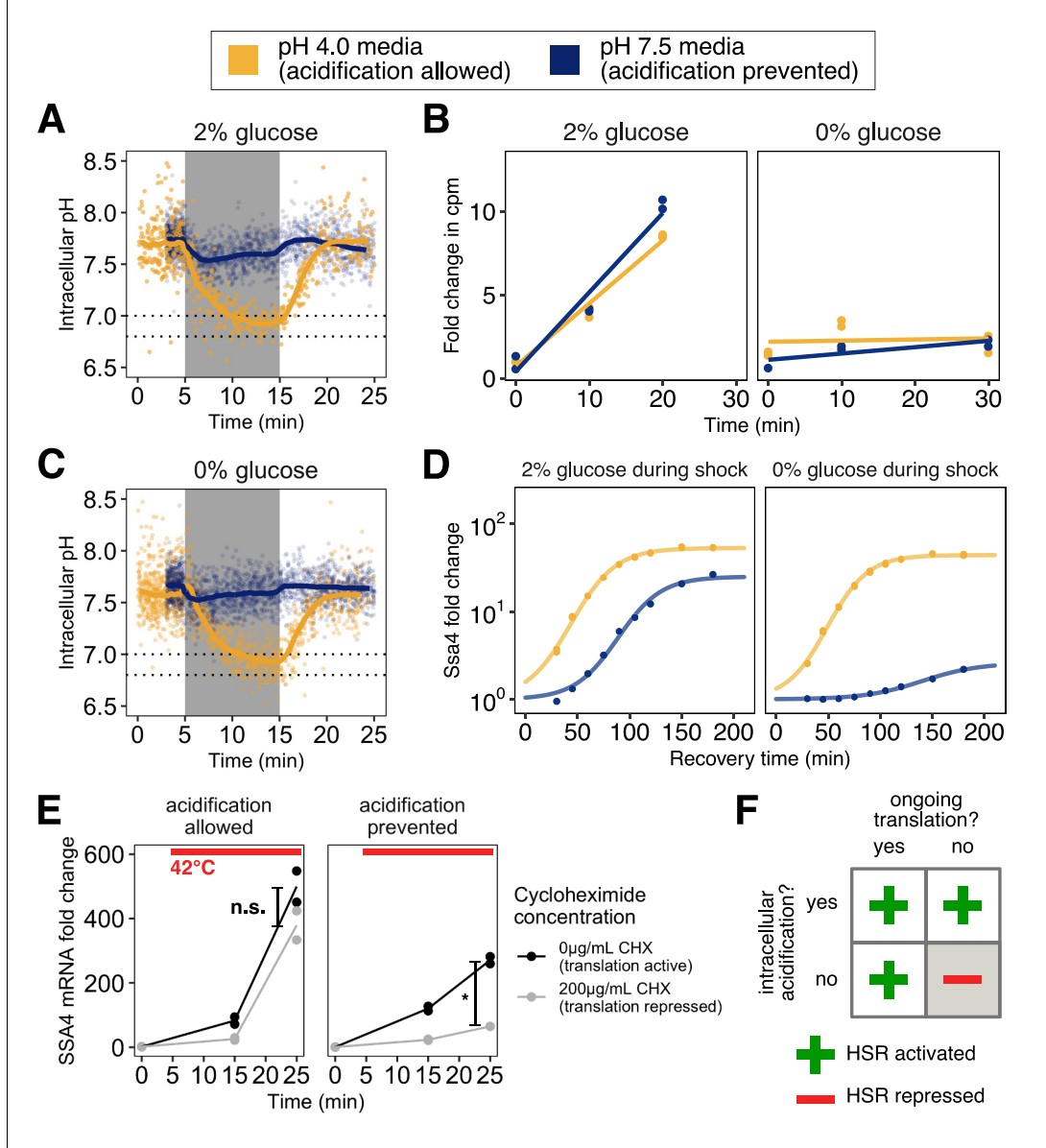

**Figure 2.** Preventing stress-associated acidification delays or impairs the heat shock response when cells are translationally inactive. (**A**) Intracellular pH changes as a function of environmental pH. Cells stressed in acidic media (pH 4.0, yellow) acidify, whereas cells stressed in media at the resting pH (7.5, blue) do not. (**B**) Inhibition of translation by glucose withdrawal does not depend on environmental pH. Incorporation of radiolabeled amino acids into total cellular proteins in counts per minute (cpm) as a function of time after a switch from medium at pH 4 with 2% glucose to the indicated media. (**C**) Same as A, but in glucose-free medium. (**D**) Induction of Ssa4 in cells able (yellow) or unable (blue) to acidify during heat shock. All measurements are of cells recovering in media containing 2% glucose. (**E**) Acidification promotes the transcriptional heat shock response (production of *SSA4* mRNA) when cells are treated with the translation inhibitor cycloheximide (200 μg/mL) prior to heat stress. *, $p = 0.017$; n.s., $p = 0.11$, Welch two-sample t-test. (**F**) Acidification promotes the heat shock response, and is required when cells are translationally inactive.

The online version of this article includes the following figure supplement(s) for figure 2:

**Figure supplement 1.** Translation and heat shock response after maltose withdrawal.

We confirmed that the sharp dependence of heat-shock protein production on intracellular pH was not due to differences in intracellular acidification resulting from translation inhibition (*Figure 2—figure supplement 1C*, compare to A; pre-shock pH change is an artifact of smoothing) or to differences in fluorophore maturation (*Figure 4—figure supplement 1A*), and that translational suppression did not depend on media pH (*Figure 2B*). To determine that the differences we saw were due

to translation state and not nutrient withdrawal, we performed the same set of experiments with cells grown in maltose, a sugar which does not cause translational suppression when rapidly withdrawn (*Ashe et al., 2000*), an effect we confirmed and demonstrated was independent of external pH (*Figure 2—figure supplement 1A*). As with cells in glucose, cells stressed in the presence of maltose were able to respond to heat shock regardless of whether they were able to acidify. Crucially, unlike glucose withdrawal, heat shock following maltose withdrawal resulted in production of Ssa4 during recovery independent of acidification (*Figure 2—figure supplement 1B*). These results indicate that ongoing translation, rather than nutrient status, predicts the cellular response.

As an additional test of the connection between translation status and the heat shock response, we treated cells with cycloheximide, an inhibitor of translation elongation, followed by heat stress where acidification was either allowed or prevented. Because cycloheximide prevented translation of the Ssa4 protein, we instead measured induction of the *SSA4* transcript using qPCR. We found that when cells were able to acidify, they robustly responded to heat shock regardless of whether they could translate. When acidification was prevented, translational repression markedly inhibited the transcriptional response (*Figure 2E*).

From these data we conclude that rapid, robust chaperone expression following heat shock depends either on ongoing translation, as previous studies have found, or on intracellular acidification (*Figure 2F*), an effect which has not been reported before. We therefore set out to determine whether this observed effect on Ssa4 extended to the broader transcriptional response.

## Failure to acidify during heat shock impairs the core transcriptional stress response regulated by Hsf1

Our results thus far link pH regulation to the translation of a limited number of heat shock proteins (*Figure 4—figure supplement 1A*). Since the heat shock response is characterized by conserved changes in transcription of multiple regulons, we used RNA-seq to characterize the stress response with and without acidification and under various translation conditions, using the pH of the media to prevent or allow acidification and treatment with 200 µg/mL cycloheximide or acute glucose withdrawal to prevent translation (*Figure 2B,E*). As controls, we assayed cells exposed to the same treatments without heat shock to account for transcriptional changes due to changes in translation and media pH. Although the results for translation arrest with both cycloheximide and glucose withdrawal are often similar, the glucose withdrawal results are more varied, and for clarity only the cycloheximide data are shown here. Equivalent versions of all figures for glucose withdrawal are shown in *Figure 3—figure supplement 1*, *Figure 3—figure supplement 2*, and *Figure 3—figure supplement 3*, and instances where the two differ substantially are noted in the text.

The transcriptome-wide response to heat shock as a function of acidification and translation state are shown in *Figure 3A* (mean of two biological replicates, see *Figure 3—figure supplement 1A* for correlation between replicates and data quality; summary shown in *Figure 3—figure supplement 2A*). As expected, actively translating cells responded by strongly upregulating heat shock genes identified by previous studies (see Materials and methods for gene annotations) independent of the pH during stress, whereas cycloheximide-treated cells showed reduced induction of heat shock genes when acidification was also blocked (*Figure 3A*; *Figure 3—figure supplement 2A*).

To further examine this apparent pH dependence of the broader transcriptional response to heat shock, we compared the heat-shocked transcriptome abundances with and without acidification against one another for both translating and non-translating populations (*Figure 3B*). Genes whose relative levels are independent of pH lie along the diagonal, genes preferentially induced in acidified cells lie above the diagonal, and genes repressed by acidification lie below. The vast majority of the heat shock genes show little pH dependence in cells stressed while translation is ongoing (*Figure 3B*, left hand side). Although this is still true for many genes during heat shock with inhibited translation, a subset of heat shock genes including *SSA4*, *BTN2*, and *HSP26* were particularly pH-sensitive. To further differentiate between induced genes and characterize the pH sensitivity of the response, we examined the transcription factors responsible for regulating the response to heat shock.

Three main transcription factors regulate yeast's heat shock response: Hsf1, which regulates chaperone-centric stress responses in all eukaryotes (*Morano et al., 2012*; *Mendillo et al., 2012*), and Msn2/4, a pair of paralogous factors limited to fungi (*Gasch et al., 2000*; *Estruch, 2000*; *Nicholls et al., 2004*). Recent work has used multiple methods to clearly define the regulons of both

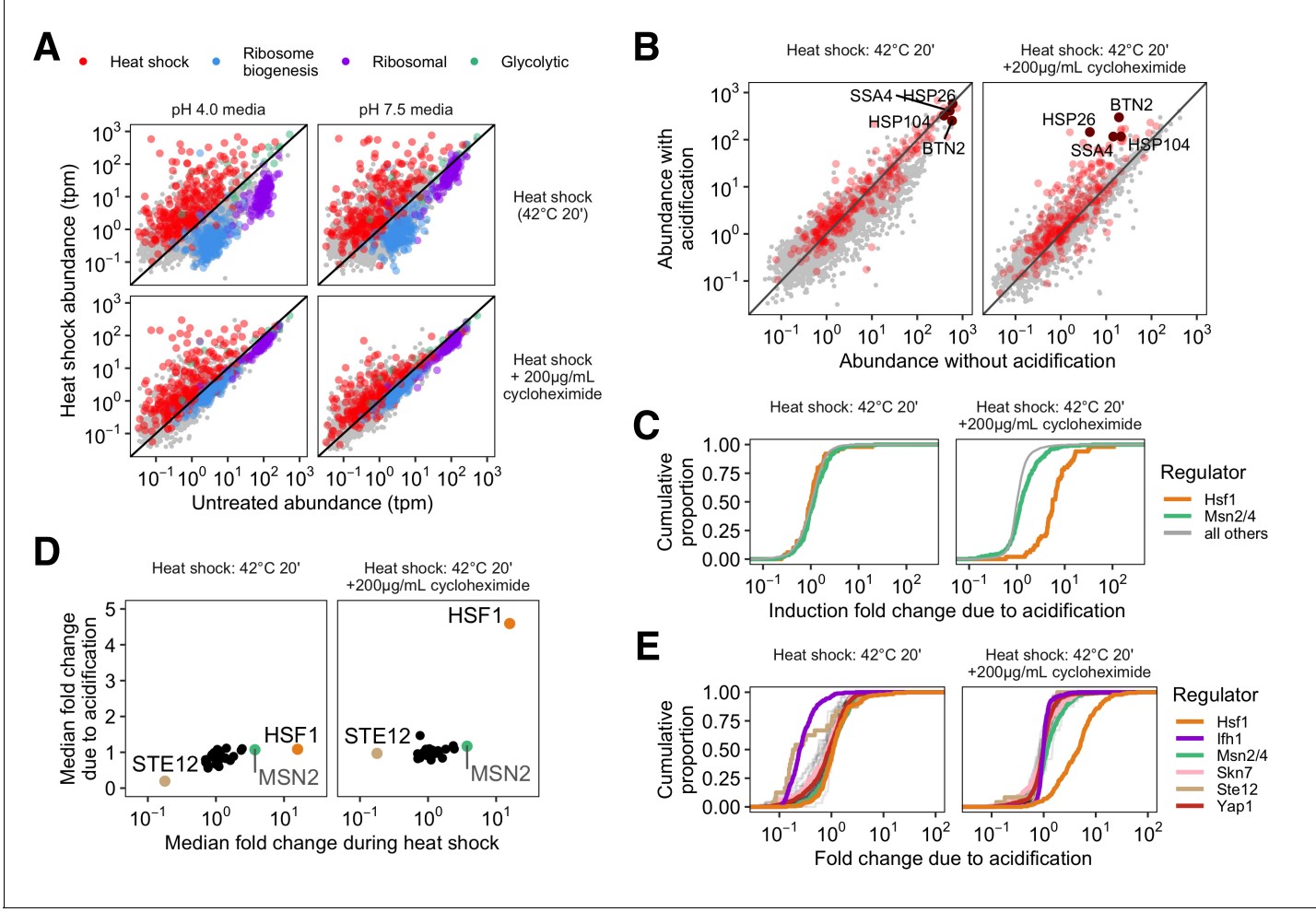

**Figure 3.** Failure to acidify during heat shock specifically represses Hsf1-activated genes. (**A**) Transcript abundance (transcripts per million, tpm) in stressed versus unstressed populations of cells. Colors correspond to gene type; gray points show uncategorized genes. (**B**) Direct comparison of gene abundance after heat shock with and without acidification. Left, data for actively translating cells; right, with translation repressed by cycloheximide treatment. (**C**) Cumulative distribution of per-gene transcript abundance in cells heat shocked with acidification relative to cells shocked without acidification (induction fold change due to acidification). Genes regulated by Msn2/4 are in green, genes regulated by Hsf1 are in orange, and all detected genes are in gray. (**D**) Mean fold change during heat shock versus the median fold change due to acidification for the regulons of all transcription factors in the YeastMine database annotated under conditions of heat stress. (**E**) Cumulative distribution of acidification fold change during heat shock for a subset of stress-involved transcription factors.

The online version of this article includes the following figure supplement(s) for figure 3:

**Figure supplement 1.** Variation between replicates and induction of heat shock genes after brief acute glucose withdrawal.

**Figure supplement 2.** Summary of fold change after heat stress by gene type and transcription factor as a function of translation state and acidification.

**Figure supplement 3.** Normalization to mock treatment and distributions of pH sensitivity for additional transcription factor regulons.

**Figure supplement 4.** Gene induction and pH sensitivity after heat stress following pH manipulation with ionophore.

(*Solís et al., 2016*; *Pincus et al., 2018*) and to identify genes regulated specifically by one or the other. We observed induction of both regulons when cells are stressed without translation, but substantially less induction of the Hsf1 regulon in cells prevented from acidifying (*Figure 3—figure supplement 2B*). To quantify the sensitivity to acidification we calculated the mRNA abundance ratio after heat shocks with and without acidification. The distributions of ratios for all genes are shown in *Figure 3C*; distributions with more genes preferentially induced in acidified cells lie further to the right. The pH sensitivity of the Hsf1 regulon is remarkably clear when cells are translationally inhibited (*Figure 3C*). Because the Msn2/4 regulon continues to induce robustly independent of pH and

translational status, a broader effect of pH on transcriptional processes cannot explain Hsf1's sensitivity.

How unusual is the Hsf1 regulon's sensitivity to pH during heat shock? We assessed the pH sensitivity of the regulons of a broad panel of transcription factors for which data are available under heat shock conditions (see Materials and methods) by comparing the mean heat-induced fold change in each regulon with and without acidification. In translationally active cells, no regulons show much acidification-dependent induction (*Figure 3D*, left). However, when we inhibited translation with cycloheximide, Hsf1 became a clear outlier, showing strong acidification-related induction (*Figure 3D*). Hsf1 pH sensitivity also emerged when translation was attenuated by glucose withdrawal, although the cellular response was more complicated overall. In line with previous results (*Zid and O'Shea, 2014*), acute glucose withdrawal alone caused some induction of heat shock genes (*Figure 3—figure supplement 3B*), however Hsf1 is still preferentially active in cells that acidify during heat shock in a way which cannot be explained by the mock treatment (*Figure 3— figure supplement 3A*). We computed the pH sensitivity of all other upregulated genes outside the Hsf1 and Msn2/4 regulons to determine whether acidification affected the global transcriptional response (*Figure 3—figure supplement 2C*). When translation is repressed by cycloheximide treatment prior to heat shock, induction of other upregulated genes is not pH-sensitive. When translation is repressed by acute glucose withdrawal, other upregulated genes show some pH sensitivity, but the Hsf1 regulon is more pH-sensitive (p $<2.2 \times 10^{-16}$, Wilcoxon rank sum test). We conclude that Hsf1's regulon stands apart in its pH-dependent induction during heat shock when translation is inhibited.

Finally, we widened our search to include other transcription factors that change their regulation during heat shock but have only been annotated under non-stress conditions; the full distributions for each regulon are shown in *Figure 3E*. When translation is arrested during heat shock, Hsf1 is the only transcription factor examined that shows significant sensitivity to acidification. Interestingly, the regulon of another transcription factor, Ifh1, was pH-sensitive only under conditions where cells were translationally active (*Figure 3E*, left hand side). The apparent pH-dependent repression of Ifh1's regulon could represent suppression of activation or a pH-sensitive mRNA decay process. Repression of the Ifh1 regulon depends on the synthesis of ribosomal proteins (*Albert et al., 2019*); we find that it also depends on intracellular acidification, but unlike Hsf1 this pH sensitivity is not dependent on translational status.

From these results, we conclude that intracellular acidification differentially affects the regulons of several transcription factors. Most strikingly, when cellular translation is halted—conditions under which classical models predict little or no Hsf1 activation—we find that intracellular acidification specifically promotes induction of genes under control of Hsf1. This highlights a previously unknown facet of the regulation of this important transcription factor. We consider potential mechanisms for this pH sensitivity in the Discussion.

## Manipulating intracellular pH during heat shock reveals the precise relationship between pH and the heat shock response

To determine the quantitative relationship between intracellular pH during heat shock and chaperone production, we sought a means to manipulate intracellular pH which would circumvent cellular regulation of the proton gradient. To accomplish this, we chemically manipulated intracellular pH using nigericin, an ionophore (*Valkonen et al., 2013*; *Triandafillou and Drummond, 2020*). Ionophores allow ions to penetrate cell membranes, temporarily destroying the electrochemical gradient. Nigericin is a $K^+/H^+$ antiporter (*Freedman, 2012*) which has been used in a variety of biological systems to equilibrate intracellular and extracellular pH (*Modi et al., 2009*; *Nakata et al., 2010*; *Thomas et al., 1979*; *Christen et al., 1982*). Importantly, this ionophore treatment was performed in the absence of glucose, conditions under which translation is halted and, as shown above, acidification strongly promotes the response to heat shock. We verified that by placing cells in buffers at different pHs and treating with ionophore we were able to accurately manipulate intracellular pH, and that this control did not depend on temperature (*Figure 4B*). We also verified that ionophore treatment alone did not have long-term fitness consequences by measuring the relative growth rate of treated and untreated cells (*Figure 4—figure supplement 1C*). Finally, we performed RNA-seq on cells heat shocked in the presence of ionophore at both the stress-associated pH (6.8) and resting

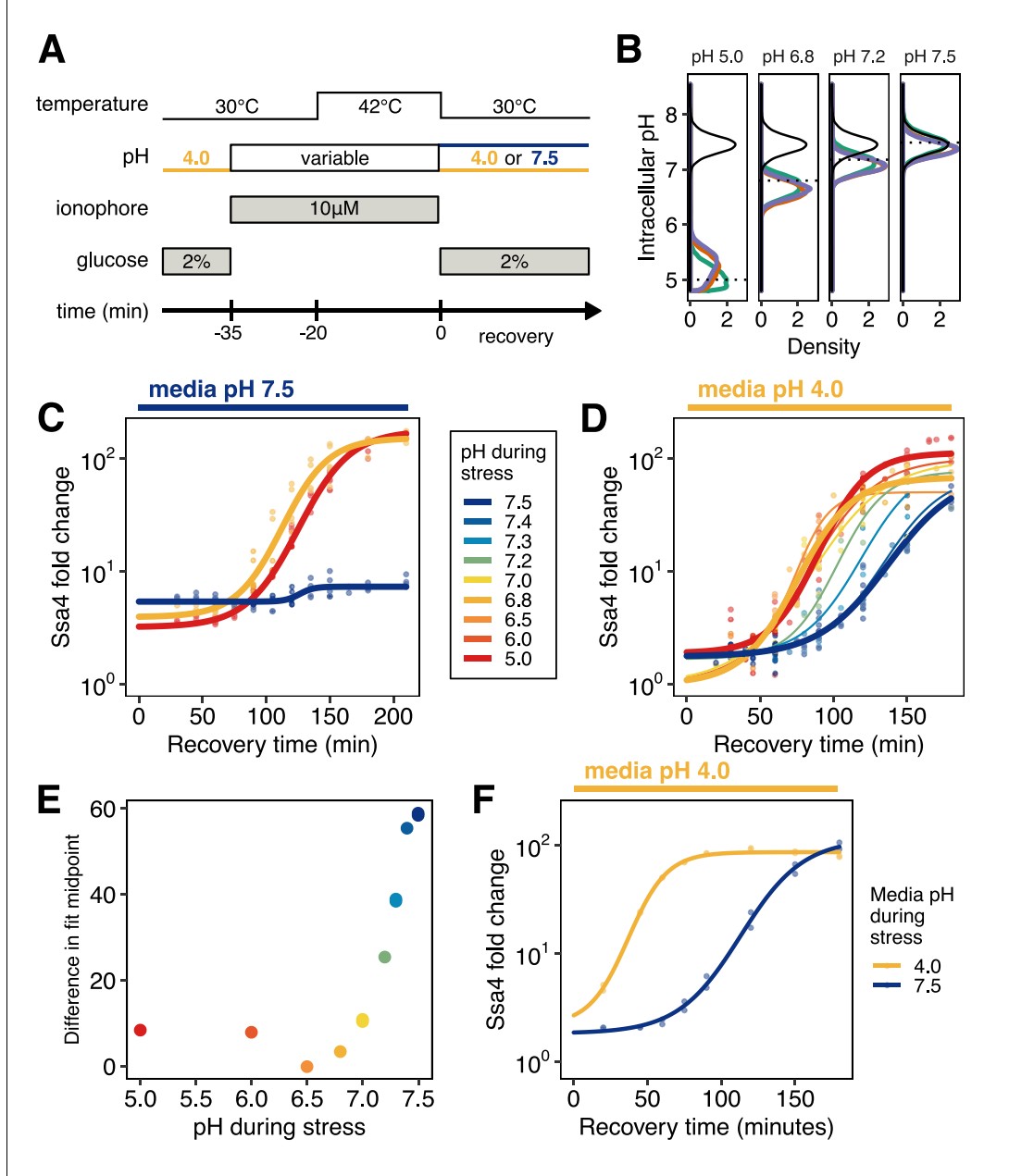

**Figure 4.** Quantitative control of intracellular pH reveals that, in the absence of translation, acidification is required for Ssa4 induction. (**A**) Schematic of intracellular pH manipulation experiments. (**B**) Intracellular pH is accurately manipulated during stress. Intracellular pH distributions were measured to determine the efficacy of pH manipulation before (green), during (red), and after (purple) 42°C heat stress. Dashed lines indicate buffer pH, and the black distribution shows unmanipulated cells for comparison. (**C**) Manipulation of intracellular pH with ionophore reproduces the acidification-dependent induction of Ssa4. Compare to *Figure 2B*, right hand side. (**D**) Fold change in Ssa4 expression following stress at different intracellular pHs and recovery in acidic media. Points represent the median of individual measurements; at least three biological replicates were performed for each condition (see Materials and methods). Lines are sigmoid fits (see Materials and methods for fitting details). (**E**) pH dependence of the induction delay; points are the midpoint of the sigmoidal fits in D. (**F**) Dependence of the stress response on media pH, followed by recovery in acidic media, recapitulates the pH dependence of the stress response when ionophore treatment is used; compare to D.

The online version of this article includes the following figure supplement(s) for figure 4:

**Figure supplement 1.** Western blot against Ssa4-mCherry and native Hsp26 after heat shock with and without acidification and stress protein production after acidification only.

pH (7.4) (*Figure 1B*) and observed the same acidification-dependence of the Hsf1 regulon under these conditions (*Figure 3—figure supplement 4*).

Using this treatment, we were able to cause cytosolic acidification without concurrent heat stress. Manipulating intracellular pH independent of temperature allowed us to determine that acidification alone was not sufficient to produce a stress response (*Figure 4—figure supplement 1D*, right hand side), with the exception of the lowest pH examined, pH 5.0, which is substantially below the range of physiologically realized pH values during short-term heat shock (*Figure 1B*).

Exposing ionophore-treated cells to heat shock (42°C for 20 min; *Figure 4A*) at a range of buffer-controlled pH levels permitted us to monitor the effect of intracellular pH on the subsequent response. After heat shock with pH control, we returned cells to ionophore-free media at 30°C and monitored Ssa4 induction by flow cytometry. Treatment with buffer and ionophore delayed the chaperone production in all samples relative to untreated cells, but did so consistently and did not affect the ultimate induction level (*Figure 4—figure supplement 1B*), supporting the assumption that pH-dependent differences between treatments can be appropriately interpreted.

Using the ionophore to manipulate intracellular pH, we were able to reproduce the same phenotype we observed in cells stressed in media with and without the ability to acidify—populations stressed at the physiological stress pH (6.8, *Figure 1*) were able to respond, and those stressed at the resting pH were not (*Figure 4C*). Furthermore, we found that additional acidification during stress, as low as pH 5, did not increase or decrease the response compared to physiological acidification.

Our initial experiments involved allowing cells to recover in media buffered to the resting pH, ensuring that the differences in the stress response were due to pH during stress. However, we noticed that proton availability after stress seemed to influence the response. Remarkably, when we heat shocked cells and prevented acidification, but allowed them to recover in acidic media, these cells were able to induce Ssa4 where cells recovering in buffered medium were not (*Figure 4D*, compare to *Figure 4C*). This recovery-media-pH-dependent induction occurred with a pH-dependent delay (*Figure 4E*) which was maximal when cells did not experience acidification during stress. To ensure that this was not due to treatment with the ionophore, we performed the same experiment without ionophore, stressing cells in media that was acidic or at the resting pH, and recovering in acidic media. The same pattern of induction was observed: cells recovering in acidic media induced Ssa4, but with a substantial delay (*Figure 4F*). What could explain the dependence on media pH during recovery for induction of the stress response? One possibility is that acidification occurs after stress and enables the induction of the stress response; we test this proposal in the following section.

We draw several conclusions from these data. The physiologically observed acidification of the cytosol is necessary for rapid heat shock protein production when translation is repressed. Physiological levels of acidification alone do not activate the response. Depriving translationally inactive cells of the opportunity to acidify virtually silences chaperone production after heat shock, an effect which is mostly transcriptional. Cells offered the chance to acidify after heat shock are still capable of mounting a response albeit with a substantial delay. All this suggests that intracellular pH during recovery plays a significant role in the production of heat shock proteins, so we turned our attention to that possibility.

## Reversal of stress-induced acidification during recovery promotes heat shock protein production in single cells

How does intracellular pH during recovery influence heat shock protein production? In acidic media, without pH manipulation, intracellular pH rapidly returns to pre-stress (resting) levels after return to ambient temperature (*Munder et al., 2016*; *Figure 1B*). We therefore wondered whether this intracellular pH recovery depended on the pH experienced during stress, and if it affected the response to heat shock. We examined intracellular pH restoration in cell populations heat shocked at different ionophore-enforced pHs and allowed to recover in acidic media. Populations stressed under acidic conditions rapidly restored intracellular pH during recovery (*Figure 5A* and *Figure 5—figure supplement 1A*). In contrast, cells stressed at pH values above 7.0 took longer on average to restore intracellular pH to resting levels, and in some cases failed to do so even after two hours (*Figure 5A*). This effect was not due to ionophore treatment; when we examined cells stressed in acidic media versus media at the resting pH, we observed the same pattern (*Figure 5B*).

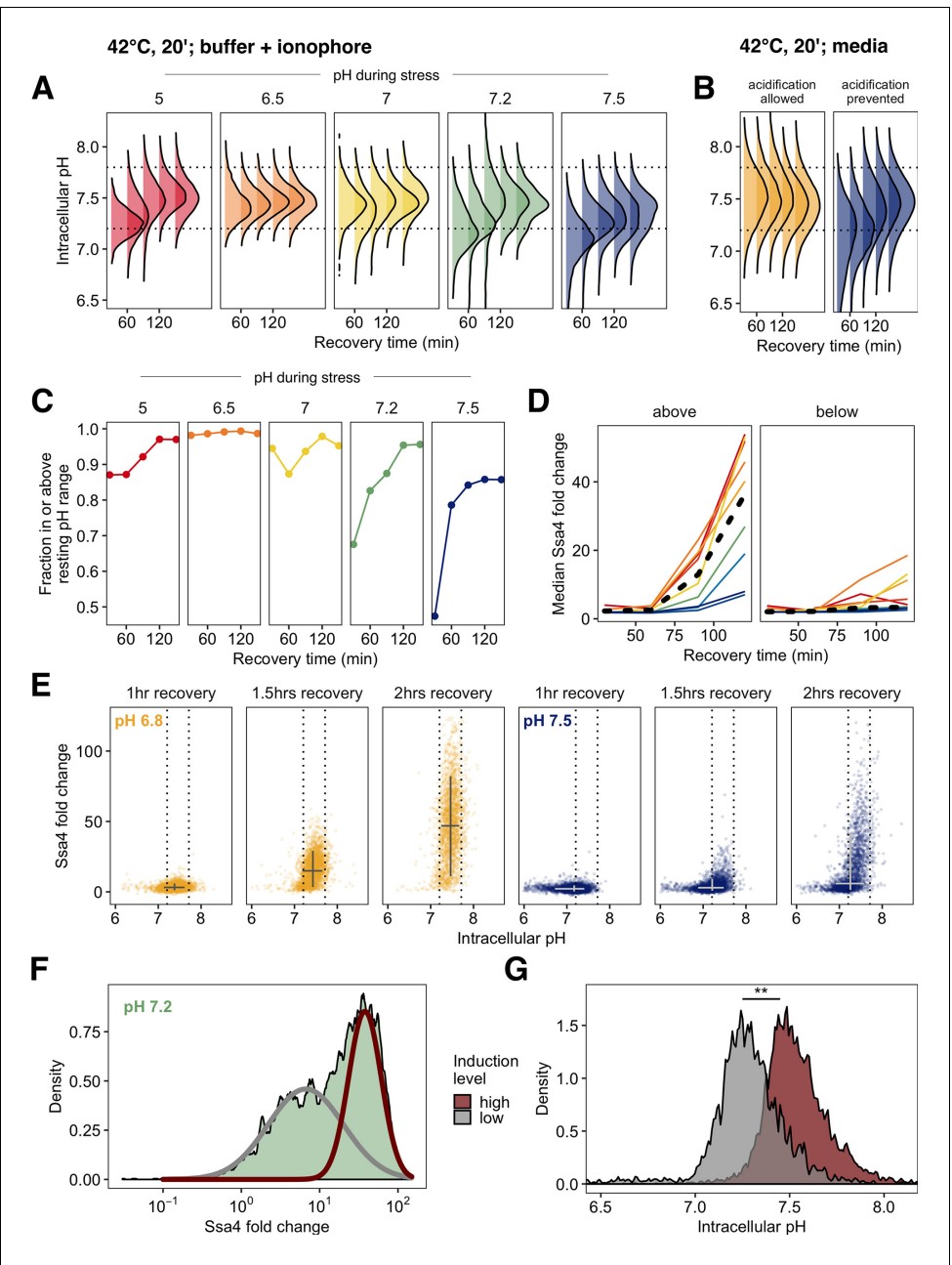

**Figure 5.** Post-stress acidification can rescue induction of the heat shock response. (**A**) Intracellular pH distributions during recovery in cells held at various pH values during 42°C heat shock. Dashed horizontal lines represent the resting pH range for untreated cells. (**B**) Same as A, but measurement made following 42°C treatment in media without ionophore. (**C**) Fraction of cells that have entered the resting pH range during recovery. (**D**) Ssa4-mCherry fold change in cells above (left) or below (right) the lower bound of the resting pH range. Color is pH during stress, black line is the median of all cells. (**E**) Relationship between intracellular pH and Ssa4 fold change on the single cell level during recovery. Return to the resting pH, bounded by dotted lines, appears to precede Ssa4 induction, and is necessary but not sufficient for high expression levels. (**F**) Distribution of Ssa4 fold change during recovery from heat stress at pH 7.2. A two-component mixture model was used to classify cells into two groups: low (gray) and high (red) induction level (>0.90 posterior probability cutoff used for assignment). (**G**) Distribution of intracellular pHs in cells belonging to either the high-expression class (red) or the low-expression class (gray). ** *P*<0.01, Wilcoxon rank sum test.

The online version of this article includes the following figure supplement(s) for figure 5:

**Figure supplement 1.** Mean pH during recovery for all pH values during stress, including averages and single-cell data.

These results support the hypothesis proposed in the previous section: cell populations held at the pre-stress pH during stress acidified during recovery. These populations—which also showed pH-dependent delays in heat shock protein production—consistently had a larger proportion of cells outside the resting pH range (*Figure 5C*). We noted that on average, cells that had failed to return to the resting pH range also failed to induce Ssa4 (*Figure 5D*). This led us to investigate the connections between intracellular pH recovery and chaperone production on the single-cell level.

Examination of the relationship between intracellular pH variation and production of Ssa4 in single cells revealed a clear pattern: virtually all cells that produced high levels of Ssa4 had returned to the resting pH (*Figure 5E*), and cells which did not return to the resting pH showed low levels of Ssa4 for up to three hours (*Figure 5—figure supplement 1B*). The vast majority of cells which had restored the resting pH after 105 min of recovery went on to robustly induce Ssa4 (*Figure 5—figure supplement 1B*). Cells far from the observed pre-stress resting pH induced less chaperone.

We further noticed that some populations showed a bimodal distribution of Ssa4 induction. In particular, we observed this behavior in populations stressed between pH 7.5 and pH 7.0. *Figure 5F* shows this distribution for cells stressed at pH 7.2; all distributions are shown in *Figure 5—figure supplement 1C*. The existence of subpopulations within identically treated samples which show different Ssa4 induction created a natural experiment, permitting us to test a strong version of the hypothesis that pH recovery is required for chaperone induction. We predicted that cells showing lower Ssa4 expression would have a lower intracellular pH compared to those with higher expression.

To test this prediction, we assigned cells to low- and high-expression categories by fitting the data with a mixture of two Gaussian distributions (*Benaglia et al., 2009*) at each timepoint (*Figure 5F*). We found that the lower-expressing subpopulation had a distinctly acid-shifted intracellular pH compared to the high-expressing cells (*Figure 5G*), confirming our prediction. Particularly at 120 min of recovery, when we see strong bimodality (*Figure 5—figure supplement 1C*), we also see strong separation of the intracellular pH distributions, with the low-expressing cells displaying intracellular pH values that fall below the ordinary unstressed range.

These data demonstrate that although cells require acidification during stress to mount a rapid response, the response further depends on subsequent reversal of acidification. Return to the resting pH predicts the dynamics of chaperone production. Acidification, either simultaneous with or following heat stress, followed by return to the resting pH appears to be required for robust induction of chaperones after heat stress.

## Precisely tuned stress-associated acidification increases cellular fitness during recovery from heat shock

In light of the connections we have established between intracellular pH changes and the induction of heat shock proteins, we sought to determine whether these pH changes promoted fitness during recovery from heat stress.

In single-celled organisms such as *S. cerevisiae*, fitness differences can be quantified by measuring the instantaneous growth rate relative to a competitor. This growth rate difference can be accurately measured by quantifying the slope of the logarithm of the ratio of population sizes as a function of time (*Geiler-Samerotte et al., 2011*). The difference in instantaneous growth rate, also known as the selection coefficient, quantifies how much better (positive) or worse (negative) cells grow relative to this reference competitor. Growth differences from two strains can then be directly compared to assess growth differences between conditions, independent of the reference.

To measure fitness differences due to acidification during stress, we heat-shocked pHluorin/Ssa4-mCherry dual-labeled cells in the presence of ionophore with a range of extracellular pH levels, enforcing a range of intracellular pH values as before. We then mixed these cultures with exponentially growing wild-type cells as the competitive reference and monitored relative proportions of these populations during recovery (*Figure 6A*). We performed additional controls to correct for potential strain differences and for the fitness effect of ionophore (see Materials and methods and *Figure 6—figure supplement 1*).

Population growth rate during recovery depended strongly on intracellular pH during heat shock. As expected, all heat-shocked populations grew more slowly than the unshocked control, with a minimum growth rate defect of −0.0043/min (*Figure 6B*), equivalent to a nearly four-fold increase in instantaneous doubling time. Maximum fitness was achieved by populations with intracellular pH

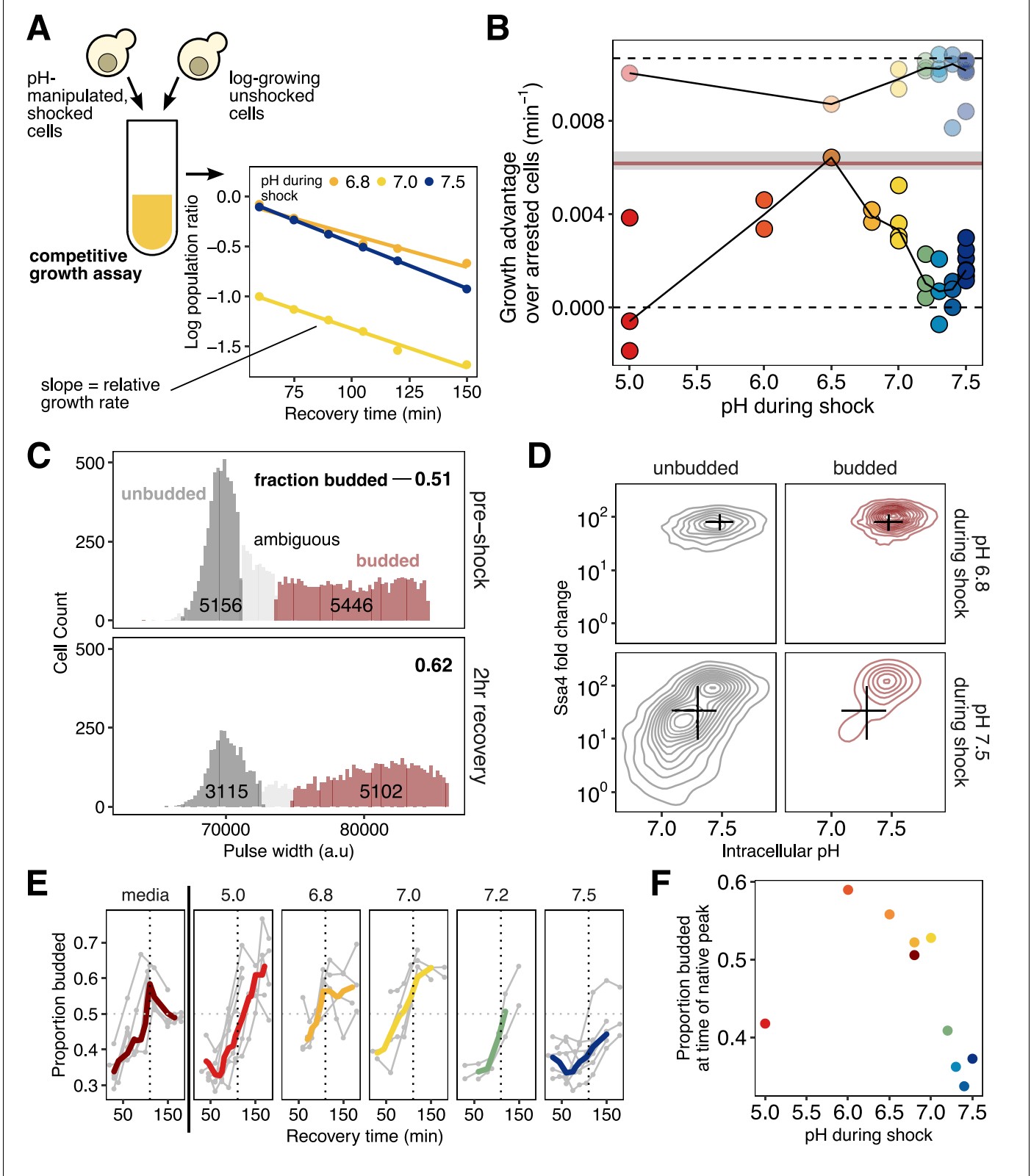

**Figure 6.** Intracellular acidification during heat shock promotes increased fitness during recovery on the population and single-cell levels. (**A**) Schematic of the competitive growth assay which measures population fitness. (**B**) Intracellular pH during heat shock vs. relative growth rate expressed as the difference from the theoretical minimum for completely arrested cells. Each point is an independent experiment; opaque points are heat-shocked populations, transparent are control populations treated with ionophore at room temperature. Gray bar is the equivalent fitness loss for cells shocked

*Figure 6 continued on next page*

*Figure 6 continued*

without pH manipulation. See Materials and methods and *Figure 6—figure supplement 1F* for details and all fits. (C) Classification of cells: large/budded (red) and small/unbudded (dark gray). Classification was performed by fitting the forward-scatter pulse width to a two-component Gaussian mixture model and using a 90% confidence cutoff to classify cells into each category; cells that did not meet this criterion (shown in light gray) are not included in the analysis. Numeric labels show the number of cells in each category. (D) Ssa4 fold-change versus intracellular pH for budded and unbudded cells during recovery at three hours post-shock. Black lines are summary statistics of the entire population (budded and unbudded) and span the middle 50% of the data, crossing at the median. (E) Proportion of cells budded as a function of time during recovery. The characteristic shape of the curve derived from cells stressed without pH manipulation is shown in the left-most panel. The proportion budded peaks at approximately two hours of recovery (vertical dashed line). (F) Summary of E, showing the average proportion of cells budded between 90 and 120 min after heat shock.

The online version of this article includes the following figure supplement(s) for figure 6:

**Figure supplement 1.** Confirmation of budding classifier by microscopy and DNA staining by flow, full fits for slopes plotted in B.

enforced to be close to its observed physiological stress-induced levels (~6.8) in unmanipulated cells (*Figure 1*). We observed the largest fitness defects in populations with pH levels set at pre-stress levels (~7.5). Ionophore treatment alone had only a minor, pH-independent effect on cell growth (*Figure 6B* and *Figure 4—figure supplement 1C*). We conclude that mimicking physiological intracellular acidification during stress maximizes fitness during recovery, consistent with acidification playing an adaptive role in the heat shock response.

We have established that differences in intracellular pH during heat shock cause differences in heat-shock protein expression at the population and single-cell levels, and that these same pH differences also cause differences in population fitness. To what extent do differences in expression cause observed differences in fitness? A causal model is motivated by the repeated observation that chaperone expression and subsequent dispersal of stress-induced aggregation precedes resumption of normal translation and progression through the cell cycle (*Cherkasov et al., 2013*; *Kroschwald et al., 2015*).

A causal, directed link from pH to chaperone expression to growth predicts that (1) cell-to-cell variation in pH will predict cell-to-cell variation in both chaperone expression and growth, and (2) cells which have resumed progression through the cell cycle will have high levels of chaperones. To test these predictions, we monitored variation in cell cycle progression and chaperone expression as a function of intracellular pH in single cells within identically treated populations.

Cellular growth and division reflect progression through the cell cycle. In budding yeast, this progress can be tracked morphologically, because emergence of a bud signals that cells have exited the gap one phase ($G_1$) and have passed through the START cell-cycle checkpoint (*Hartwell, 1974*). Heat stress causes yeast cells to arrest in $G_1$, an unbudded state (*Rowley et al., 1993*). The presence of a bud following stress indicates that the cell has re-entered the cell cycle and begun reproducing. Cells without a bud cannot be confidently assigned to a growth phase because they may either be arrested in $G_1$ or merely passing through this phase during active growth. To classify budded cells, we adapted a microscopy-based assay (*Rowley et al., 1993*) for flow cytometry, using the pulse width of the forward scatter channel to measure cell (or cell plus bud) length. From the resulting data we scored cells as budded, unbudded, or ambiguous (*Figure 6C*); see Materials and methods and *Figure 6—figure supplement 1* for full details (*Tzur et al., 2011*; *Hoffman, 2009*; *Hewitt and Nebe-Von-Caron, 2004*). Within these morphological categories, we then could assess the relationships between cell-cycle state, intracellular pH, and chaperone expression.

Cells heat-shocked at pH 6.8, mimicking normal acidification, showed robust chaperone expression during recovery. In contrast, cells shocked at pH 7.5, preventing acidification, created a large subpopulation of cells in which chaperone expression was suppressed (*Figure 6D*). Virtually all cells which could be confidently assigned to the budded state showed high chaperone expression, and nearly all cells showing low chaperone expression were found in the unbudded state. Low-expression, unbudded cells also showed near-uniform reduction in pH (*Figure 6D*, lower left panel), consistent with the dysregulation of pH observed at the population level. These observations match both above predictions of a causal relationship between chaperone expression and cell-cycle resumption, modulated by intracellular pH.

We also used the proportion of budded and unbudded cells as an orthogonal measure of population-level fitness, measuring resumption of the cell cycle as opposed to cell doubling. During recovery, cells are released from heat-induced $G_1$ arrest en masse, leading to a temporary synchronization

of the population with a coordinated increase in the proportion of budded cells, which eventually returns to the steady-state value for exponentially growing cells (*Rowley et al., 1993*) as seen in *Figure 6E* (left, dark red trace). After a 42°C, 20 min heat shock without pH manipulation, the percentage of budded cells peaked just before two hours of recovery (dashed line in *Figure 6E*). In pH-manipulated cells, if the pH experienced with elevated temperature was close to the native stress-associated pH, this recovery peak occurred at approximately the same time as in unmanipulated cells. However, cells that experienced a more acidic or more basic pH during heat shock showed a delay in the occurrence of the budding peak (*Figure 6E*, summarized in *Figure 6F*), in agreement with the difference in growth rates shown in *Figure 6B*.

By measuring growth in multiple ways, we have shown that post-stress resumption of growth is tuned to particular stress-associated cytoplasmic pH values. Moreover, fitness positively correlates both with increased chaperone production and with restoration of the pre-stress pH in populations and in individual cells. Resumption of growth is consistent, at the population and single-cell level, with induced chaperones contributing to release of stress-induced cell-cycle arrest as others have reported (*Kroschwald et al., 2015*).

## Discussion

What is the physiological significance of the broadly conserved, transient intracellular acidification triggered by stress in eukaryotes? By decoupling changes in intracellular pH from heat shock in budding yeast, we have discovered that the canonical transcriptional stress response mediated by heat shock factor 1 (Hsf1) depends on cellular acidification. When cells are translationally suppressed, such as following glucose withdrawal, transient acidification becomes a requirement for achieving a robust transcriptional response. Even in translationally active cells, acidification promotes induction. Restoration of resting pH and chaperone protein expression increase competitive fitness by promoting reentry into the cell cycle and overall population growth rates, indicating that transient acidification is an adaptive component of the heat shock response.

Our initial results are consistent with the longstanding view that misfolding of newly synthesized polypeptides can serve as Hsf1 inducers (*Baler et al., 1992*; *Masser et al., 2019*), presumably through recruitment of Hsp70 away from its repressive association with Hsf1 (*Zheng et al., 2016*; *Krakowiak et al., 2018*; *Li et al., 2017*). However, we have discovered an alternative activation pathway for Hsf1 under conditions when newly synthesized proteins are in short supply—when translational activity is low, such as following starvation or pharmacological inhibition. Here, intracellular pH plays a decisive causal role in Hsf1 activation following heat shock. Either ongoing translation or intracellular acidification is required, and the absence of either signal leads to suppression of the Hsf1-mediated transcriptional response during heat shock (*Figure 7*).

What is the source of the protons required for adaptive acidification? Our results strongly indicate that extracellular protons entering the cell following heat shock are the dominant cause of acidification. Simply placing translationally inactive cells in medium buffered to the resting cellular pH is sufficient to suppress the heat shock response during an otherwise robust heat shock, suggesting that no intracellular store of protons is liberated to cause acidification. Membrane permeability to small molecules increases with temperature in *S. cerevisiae* (*Coote et al., 1994*), and proton permeability specifically has been shown to increase with temperature (*van de Vossenberg et al., 1999*), providing a likely mechanism for temperature-dependent acidification when a plasma-membrane-spanning proton gradient is present. (We discuss below certain physiological scenarios in which an ample source of extracellular protons and heat shock will reliably co-occur.)

Our results indicate a close causal connection between intracellular pH, chaperone production, and cellular growth. A surprising yet consistent detail is that cells must restore their resting pH before producing high levels of molecular chaperones. Previous work has demonstrated that heat shock causes changes in intracellular pH (*Weitzel et al., 1985*) and that intracellular pH controls growth rate (*Orij et al., 2012*). Our results are consistent with these findings, while adding critical steps, such as demonstrating that chaperone production sits between pH and growth in the causal chain, and that these dynamics can be seen at the single-cell level.

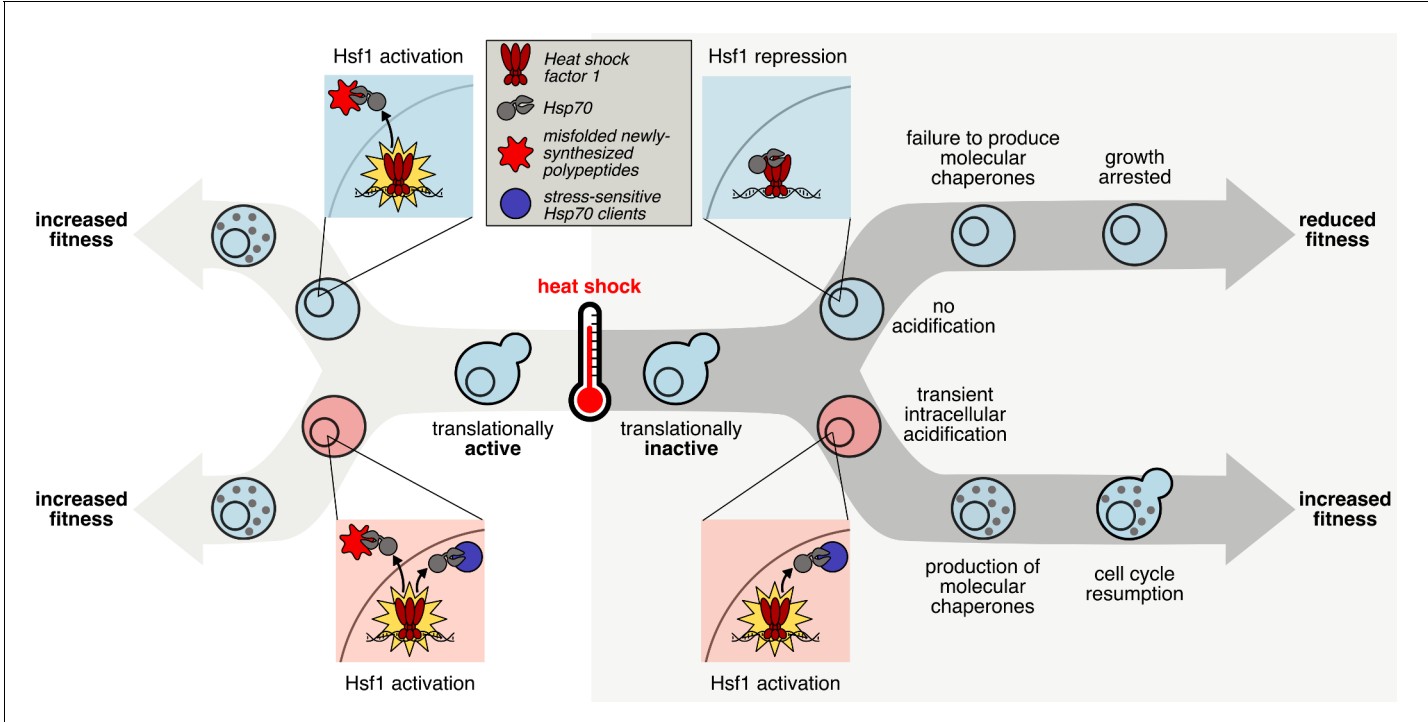

**Figure 7.** The transcriptional response to heat shock, chaperone production, and cellular fitness are promoted by intracellular acidification. The key step in initiating the transcriptional heat shock response is the release of Hsf1 repression by Hsp70 through titration of the chaperone via client binding. When cells are translationally active (left hand side), newly synthesized polypeptides that misfold in response to elevated temperature can act as the trigger. However, cells that are not actively translating (right hand side) can still respond to heat shock, dependent on transient intracellular acidification, either during or after the temperature increase. We predict that pH-sensitive, stress-sensing proteins, similar to those already discovered, can act to titrate Hsp70, relieving Hsf1 repression and activating the transcriptional heat shock response.

## The pH dependence of Hsf1 points to pH-dependent substrates of Hsf1's repressor, Hsp70

Why is acidification required to mount the transcriptional heat shock response under certain conditions? Acidification in the absence of heat shock, at least to pH levels which would normally follow heat shock, is insufficient to activate Hsf1, ruling out direct sensing of pH by Hsf1 or its repressor Hsp70. On the flip side, Hsf1 can be robustly activated without a drop in pH, so long as cells are translationally active, indicating that acidification is not necessary for Hsf1 activation. Recent key studies have demonstrated that production of Hsp70 binding substrates that titrate Hsp70 away from Hsf1 suffices to induce Hsf1 in the absence of heat shock (*Zheng et al., 2016*; *Krakowiak et al., 2018*). All these results are consistent with the standard misfolding model: newly synthesized polypeptides misfold in response to heat shock, leading to recruitment of Hsp70, which causes Hsf1 activation. The pressing question is how Hsf1 is activated in the absence of newly synthesized polypeptides. Previous results argue against widespread heat-induced misfolding of mature endogenous proteins in vivo (*Wallace et al., 2015*). Here, consistent with those results, cells show marked repression of the heat shock response at 42°C when translation is attenuated and resting pH is maintained, suggesting that misfolding caused by temperature, if it occurs, is insufficient to trigger the Hsf1 response. The remaining possibility is that Hsp70 substrates can be produced without ongoing translation in an acidification-dependent manner.

That is, we seek a mature protein which, in a heat- *and* pH-dependent manner, changes its state in a way which recruits Hsp70. Remarkably, a profusion of such candidates exists. Nearly two hundred mature endogenous proteins in yeast form reversible condensates in response to heat shock which under extreme stress coalesce into stress granules (*Wallace et al., 2015*). Hsp70 colocalizes with stress granules (*Cherkasov et al., 2013*; *Walters et al., 2015*), and stress-granule dispersal depends on Hsp70 and other chaperones (*Cherkasov et al., 2013*; *Walters et al., 2015*;

*Kroschwald et al., 2015*; *Kroschwald et al., 2018*). Three individual protein components of stress granules, poly(A)-binding protein Pab1, poly(U)-binding protein Pub1, and the DEAD-box RNA helicase Ded1, condense by phase separation in vitro when exposed to heat in a pH-dependent manner (*Riback et al., 2017*; *Kroschwald et al., 2018*; *Iserman et al., 2020*). Pab1 and Ded1 condense by phase separation which depends strongly on temperature and pH. For example, a 42°C heat shock is insufficient to cause Pab1 condensation at pH 7.5, but sufficient at pH 6.6 (*Riback et al., 2017*). In the case of Pub1, acidic pH promotes phase separation at non-shock temperatures, but these condensates spontaneously disperse when the resting pH is restored; only heat-triggered Pub1 condensates require dispersal by molecular chaperones (*Kroschwald et al., 2018*).

Together, these studies indicate the existence of multiple proteins that undergo heat-triggered, pH-dependent condensation processes, producing assemblies which conditionally recruit Hsp70. Based on these observations, we make three hypotheses to be tested in future work. First, we hypothesize that pH- and temperature-dependent condensing proteins are the cause of the Hsf1 pH-dependence we observe when translation is silenced. Second, we predict that if such proteins are found, they will activate Hsf1 by recruiting Hsp70 in the condensed state. Third, we hypothesize that many such proteins exist, such that it should be possible to activate Hsf1 by triggering condensation of a single protein, but that suppression of any single protein's condensation may not suppress Hsf1 activation.

## pH dependence constrains the search for temperature sensors in eukaryotes

How eukaryotic cells sense temperature remains unknown (*Yoo et al., 2019*). In the misfolding model for Hsf1 activation, misfolded proteins are the actors which convert an increase in temperature into Hsp70 recruitment and thereby activate Hsf1 (*Morano et al., 2012*); neither Hsf1 nor Hsp70 has temperature sensitivity in this model. Recent work has suggested that human Hsf1 possesses intrinsic thermal sensing ability regulating its trimerization (*Hentze et al., 2016*). While this is a tantalizing possibility, Hsf1 thermosensing has yet to be demonstrated in vivo or for yeast, whose Hsf1 is thought to be constitutively trimerized (*Morano et al., 2012*). Our results also demonstrate that temperature alone is insufficient to activate Hsf1's response when translation is attenuated; a drop in intracellular pH is required.

We have previously proposed that heat-triggered protein condensation can take the place of misfolding-induced aggregation in the standard model for Hsf1 activation, with phase-separating proteins acting as the primary sensors of temperature (*Riback et al., 2017*). Phase separation and other phase-transition behaviors provide a compelling solution to the tricky problem of sensing temperature, which typically involves only a few degrees' change: 30°C to 37°C for robust induction of yeast's heat shock response (*Gasch et al., 2000*). Phase transitions by definition are highly cooperative, amplifying tiny changes in individual molecules into massive system-level transformations (*Yoo et al., 2019*). Crucially, unlike misfolding of newly synthesized polypeptides, heat-shock-triggered condensation of mature proteins is not suppressed by translational inhibition (*Wallace et al., 2015*).

Which proteins might serve at the front line of temperature sensing, transducing slight temperature shifts into a cellular signal capable of triggering the Hsf1-mediated heat shock response? We have previously identified more than a dozen proteins which form condensates in under two minutes in response to heat shock in vivo (*Wallace et al., 2015*). These so-called 'superaggregators' condense rapidly; many of them reside in the nucleus; and most show substantial condensation at 37°C, unlike Pab1 or Pub1 but like the exquisitely temperature-sensitive Ded1 (*Iserman et al., 2020*). Such sensitivity is essential for any protein acting as an initial sensor of the Hsf1-mediated response. As noted above, these thermosensitive proteins provide a compelling list of candidates for Hsf1 activators. We predict they will have several characteristics shared by existing less-sensitive proteins: they will condense autonomously (without relying on the temperature-sensitive condensation of other factors), will recruit Hsp70 upon condensation, and will show condensation behavior that is suppressed at the resting intracellular pH.

We underscore that these are predictions based on a synthesis of existing knowledge, and that the mechanistic basis for the pH sensitivity we report is now a crucial open question. It remains possible, for example, that Hsf1 directly senses pH and translational activity. A conceptual advantage of our model is that, unlike this example, it is built from empirically extant pieces.

## Temperature acts as a physiological signal

Is Hsf1 activation a response to a heat-induced proteotoxic misfolding catastrophe, or something else? Heat-induced misfolding has long remained more a supposition than a result. While it is clear that artificially induced misfolded proteins can induce the heat shock response (*Geiler-Samerotte et al., 2011*; *Trotter et al., 2002*), this does not constitute evidence that they serve as inducers under physiological conditions. As noted above, no specific endogenous protein has yet been identified which misfolds in response to a sublethal heat shock and thereby triggers the Hsf1 response. Here, we have shown that heat alone is insufficient to trigger the Hsf1 response, and that the newly synthesized polypeptides often cited as the primary inducers of Hsf1 are not required for Hsf1 activation.

An alternative to the misfolding model is that elevated temperature—within the physiological range to which organisms have adapted during their evolution—serves a signal, an environmental cue, which elicits an appropriate response.

Temperature acts as a physiological signal in other ascomycete fungi. For example, some dimorphic fungi live and grow in the environment as a mold, and convert into a yeast (a single-celled, reproducing fungus) in response to entering a mammalian host and detecting the resulting increase in temperature, the critical sensory cue (*Klein and Tebbets, 2007*). The budding yeast and occasional human pathogen *Candida albicans* similarly requires a temperature increase to trigger the bud-to-hyphae transition critical for infection (*Brown et al., 2010*), which also induces chaperones in a classical Hsf1-mediated heat shock response (*Nicholls et al., 2009*).

The foregoing examples are pathogens. What physiological event would prompt the execution of such a heat shock program in non-pathogenic *Saccharomyces cerevisiae*? *S. cerevisiae* does not produce fruiting bodies and depends upon animal hosts for dispersal (*Mortimer and Polsinelli, 1999*). This, along with other facts which we review here, suggests that a primary physiological heat shock for budding yeast is ingestion and dispersal by birds.

A survey of hundreds of migratory passerine (perching) birds (*Francesca et al., 2012*) isolated yeast species from their cloacae, implying ingestion as the source; 14% of isolates were *Saccharomyces cerevisiae*. *S. cerevisiae* survived experimental passage through birds when inoculated in feed (*Francesca et al., 2012*). Passerine birds, the most numerous on earth, have an internal body temperature averaging 41.6°C (range 39°C to 44°C) when active, rising to an average of nearly 44°C (43.1–47.7) during high activity such as running and flight (*Prinzinger et al., 1991*). These temperatures correspond remarkably well to the upper bound of nonlethal temperatures for *S. cerevisiae* (*Salvadó et al., 2011*). Ingestion will reliably induce a sudden thermal shift. The acidity of the stomach provides an ample source of protons to drive intracellular acidification.

A prominent ecological niche for *S. cerevisiae* is the surface of fruits such as grapes (*Mortimer and Polsinelli, 1999*), which birds eat—indeed, vineyard crop damage by passerine birds is a major challenge for the wine industry (*Somers and Morris, 2002*; *DeHaven and Hothem, 1981*). Yeast proliferate to higher numbers on damaged fruit (*Mortimer and Polsinelli, 1999*) which often results from bird pecking (*Chiurazzi et al., 2010*; *Somers and Morris, 2002*). Besides birds, other known dispersing hosts for the *Saccharomyces* genus include wasps, bees, ants, and fruit flies (*Dapporto et al., 2016*; *Madden et al., 2018*; *Mortimer and Polsinelli, 1999*; *Giglioli, 1897*; *Christiaens et al., 2014*), all of which are preyed upon by birds, indicating that yeast may enter an avian carrier by multiple routes. Yeast cells that survive passage through a bird stand to benefit from broad geographic dispersal, an evolutionary advantage.

From these diverse and convergent lines of evidence, we conclude that ingestion and dispersal by birds is an ecologically established, physiologically relevant, and likely evolutionarily advantageous heat-shock condition for budding yeast. To obtain this advantage, yeast must travel through an acidic, low-nutrient environment averaging approximately 42°C.

## Broader considerations

Recognition that a rise in temperature may represent a signal rather than merely a damaging agent alters how one thinks about the purpose of the response to temperature, the response's molecular triggers, and the physiological conditions under which the response would be deployed. Here, the suppression of the heat shock response by elevated pH suggests that acidification—and the capacity to acidify, which appears to be determined in large part by extracellular pH—is a key part of the

physiological context in which this thermal signal is received. This logic applies broadly. In humans, for example, a key physiological heat shock—fever—triggers the Hsf1-mediated heat shock response (*Singh and Hasday, 2013*). Perhaps fever causes new problems for cells, new self-inflicted damage to be cleaned up. More plausibly, however, fever acts as a systemic signal which activates a cellular program with key roles in modulating immune and inflammatory responses (*Singh and Hasday, 2013*). Indeed, the apoptotic response of human neutrophils to fever temperatures is sharply dependent on intracellular pH, with acidification promoting survival; local acidification is a hallmark of inflammatory conditions and promotes neutrophil activation (*Díaz et al., 2016*).

We began by noting that the biological meaning of the longstanding association of cellular stress with cytosolic acidification, observed from fungal cells to vertebrate neurons, has remained unclear. Our results speak to a potentially broad effect: that this association is adaptive, and reflects, at least in part, the dependence of the core Hsf1-mediated transcriptional response on pH. Our work will focus a decades-long search for the specific eukaryotic sensors of heat shock on systems—likely, we argue, specific molecules—which depend on acidification for their sensory action.

## Materials and methods

### Yeast strains

Scarless tagging of the Ssa4 protein with mCherry was accomplished in the BY4742 background via serial transformation and fluorophore exchange with the *URA3* gene such that no selection cassette remained in the genome. This was done by creating an intermediate strain with *URA3* at the C terminus of the *SSA4* locus; this sequence was replaced with mCherry and counterselection was done on 5-fluoro-orotic acid (5-FOA). The final strain has the *SSA4* gene in the native context with the native stop codon replaced by the mCherry sequence. In the BY4741 background, the coding sequence for pHluorin, under control of the constitutive *GPD1* promoter, was incorporated at the *LEU2* locus using Leu2 expression as a selectable marker. Strains were purified at least twice by streaking and picking single colonies, before being mated. The resulting strain, yCGT028 (MATa/α ura3Δ0/ura3Δ0 leu2Δ0/LEU2::pHluorin his3Δ0/his3Δ0 MET15/met15Δ0 lys2Δ0/LYS2::SSA4/SSA4-mCherry) was used for all experiments except those shown in *Figure 4—figure supplement 1A*, which uses strain yCGT032.

Strain yCGT032 was constructed in the same fashion, but with *SSA4* fused to a FLAG tag rather than mCherry.

### Growth and stress conditions

Unless otherwise stated, yeast cells were grown at 30˚C in synthetic complete media with 2% glucose (SCD) at pH 4. Under these conditions the doubling time of diploid cells was approximately 70 min. For all experiments, cultures were started from the same frozen stock, and grown so that the cell density was below optical density (OD) 0.1 for at least 12 hr before stress; a dilution of no more than 20-fold was performed at least 4 hr prior to stress. Cells were grown to between OD 0.05 and OD 0.1 (flow cytometry) or to OD 0.3–0.4 (RNA-seq) before being stressed.

All temperature stresses occurred at 42˚C for 20 min, except for the data in *Figure 1D* and *Figure 1—figure supplement 1C*, which are 42˚C for 10 min.

### Measuring translation rate

Yeast cells were grown at 30˚C with 250 rpm shaking in synthetic complete media with 2% glucose (SCD) for glucose withdrawal experiments or in YP + 2% maltose for maltose withdrawal experiments. Cells were grown to an OD600 of 0.2–0.3, then transferred to media adjusted to either acidic pH (four for SC, 6.5 for YP) or to the resting pH (7.5), with or without 2% sugar, and containing $^{35}$S-L-methionine and $^{35}$S-L-cysteine at a final concentration of 1 µCi/mL. Cells were grown at room temperature with no shaking (to emulate pre-stress conditions for all heat shock experiments), and aliquots were taken as a function of time. Proteins were precipitated by addition of 50% trichloroacetic acid (TCA) to a final concentration of 8.33%. Samples were placed on ice for 10 min, held at 70˚C for 20 min, then returned to ice for another 10 min before being spotted onto glass microfiber filters. Samples were washed with 5% TCA, 95% ethanol, dried at room temperature for at least 24 hr, then placed in scintillation fluid. Radioactivity was quantified by liquid scintillation counting.

## Flow cytometry

### Technical information

Two cytometers were used to collect data: BD Biosciences LSRFortessa and BD Biosciences LSRFortessa-HTS. Both were equipped with 405, 488, 561, and 620 nm lasers. Cells were run on the lowest flow rate possible. Voltage and filter sets used were as follows (two filter sets were used on the HTS instrument):

| Channel name | Fluorophore | Fortessa HTS (1) | Fortessa | Fortessa HTS (2) |
|---|---|---|---|---|
| Forward Scatter (488) | NA | 302 | 110 | 302 |
| Side Scatter (488) | NA | 242 | 236 | 236 |
| PE Texas Red (561:610/20) | mCherry | 550 | | |
| FITC (488:525/50) | pHluorin 488 | 450 | 422 | 422 |
| BV421 (405:450/50) | NA | 300 | 495 | 400 |
| BV510 (405:525/50) | pHluorin 405 | 400 | 400 | 400 |
| PEDazzle (561:610/20) | mCherry | | 625 | 625 |

All individual experiments were performed with the same voltage set, and the fluorescence values reported are normalized to a within-experiment fluorescence baseline (unstressed cells), allowing for direct comparison between experiments taken on different instruments or with different voltage sets.

Unstressed cells were used to determine manual gates on forward and side scatter to isolate cells. Growth conditions (see above section) were such that no significant populations of dead cells were expected. In some experiments a sub-population of cells became highly fluorescent in the BV421 channel. These cells were ambiguously bright in the FITC (488) channel, meaning that they could not be confidently assigned to either strain; although recorded, these cells were excluded from the analysis computationally by threshold gating in the BV421 channel. The percentage of these cells of the total initially gated population was between 5% and 50%, and varied primarily with handling (no association with pH).

### Dynamic intracellular pH measurements

Cells constitutively expressing pHluorin in the cytoplasm (yCGT028) were grown as described in Growth Conditions above. A 400 µL aliquot of cells was loaded onto the flow cytometer at room temperature and the instrument was run continuously for 5 min of equilibration. With the instrument still running, the sample tube was briefly removed and 1 mL of media at 44°C was added (to account for heat loss in mixing); the tube was rapidly returned to the cytometer and held in a 42°C water bath for 10 min, followed by 10 min at 30°C.

### Sample size and reproducibility

All flow cytometry stress experiments were performed at least in triplicate, with at least 10000 total events (cells) collected at each timepoint. Due to variation among partitioning between populations, the number of events for each sub-category varied, but was never below 1000 cells. All flow cytometry mock experiments were performed at least in duplicate, with the same standard for number of events as stress experiments.

## pH manipulation

### Calibration curve buffer

50 mM NaCl, 50 mM KCl, 50 mM MES, 50 mM HEPES, 100 mM ammonium acetate, 10 mM 2-deoxyglucose; pH adjusted with HCl or KOH. 10 mM (1000x) nigericin in 95% EtOH was added just before buffer use to a final concentration of 10 µM.

### pHluorin calibration curve

We used a protocol modified from *Valkonen et al., 2013*. Exponentially growing cells (OD 0.05–0.15) were spun out of SC media (3000 g for 2–4 min) and resuspended in calibration curve buffer at

0.5 pH unit intervals between pH 4.5 and pH 8.5. Cells were equilibrated in buffer at room temperature for 15–30 min and then analyzed by flow cytometry. The calibration curve was generated by taking the median ratio of fluorescence in the 405:525/50 (BV510, pHluorin 405) channel to the 488:525/50 (FITC, pHluorin 488) channel, and fitting the resulting points to a sigmoid:

$$ratio_{405:488} \equiv R = \frac{a}{1 + \exp(-b(pH - c))} + d \tag{1}$$

where a, b, c, and d are fitting parameters. Ratios were corrected for background by subtracting the autofluorescence of unlabeled (wild type) cells in either media (for samples in media) or buffer (for the calibration curve). A new calibration curve was measured each time an experiment was performed. A representative calibration curve is shown in *Figure 1—figure supplement 1B*. A comparison between calibration curves in shown in *Figure 1—figure supplement 1A*: although the absolute value of the ratios may vary, the calculated effective pKa of the fluorophore is expected to be consistent across experiments. The effective pKa was calculated using the formula (*Bagar et al., 2009*):

$$\log\left(\frac{R - R_{max}}{R_{min} - R}\right) = 0 \tag{2}$$

## Determining ionophore efficacy at increased temperature

To ensure that the ionophore treatment was effective at elevated temperature, the intracellular pH of cells in calibration curve buffer at 42℃ was assessed. Cells were resuspended (at the same ratio of cells:buffer as used in pH manipulation experiments) in calibration curve buffer of varying pH and equilibrated at room temperature for 15 min. A small volume was used such that thermal changes rapidly equilibrated. A portion of the cells were analyzed by flow cytometry, and then the remaining samples were placed in a heat block at 42℃. Aliquots were taken at 10 and 20 min and analyzed by flow cytometry. The intracellular pH was calculated using a calibration curve generated at 30℃ using different buffers. The close correspondence between the measured buffer pH and the calculated intracellular pH from the calibration curve is shown in *Figure 4B*.

## Manipulating intracellular pH during stress

Intracellular pH during stress was manipulated using calibration curve buffer. The concentration of the ionophore was low enough that any anti-biotic effects were negligible, as seen by the small fitness effect on pH-manipulated, unstressed cells (see *Figure 6—figure supplement 1D*, 'RT (mock)').

1.2 mL of cells grown as described in the above 'Growth and stress conditions' section were spun out of media and resuspended in 60 µL freshly prepared calibration curve buffer plus ionophore at the desired pH, equilibrated at room temperature for 15–30 min, and then either exposed to 42℃ temperature ('heat shock') or room temperature ('mock') for 20 min. After stress, cells were recovered by removing the buffer and resuspending in 1.2 mL of fresh SC media and holding at 30℃ with 250 rpm shaking. The fresh SC was either not pH-adjusted (with a pH of approximately 4, data shown in *Figure 4D*, or was buffered to pH 7.4 using 0.1 M Na$_2$HPO$_4$ : NaH$_2$PO$_4$ buffer [data in *Figure 4C*]).

### Western blotting

yCGT032 was grown in 180 mL SC media at 30℃ shaking at 250 rpm for 12 hr then harvested at OD 0.026. Three aliquots of 50 mL cells were harvested by spinning at 3000 g for 5 min. Each aliquot was washed with water and then resuspended in 1 mL of a different medium: SC, pH 6.8 calibration curve buffer with ionophore, or pH 7.4 calibration curve buffer with ionophore. Cells were equilibrated for 15 min at room temperature and then split into two samples, one for heat shock and one for mock treatment. Heat shock was performed by incubating cells in 42℃ water bath for 20 min. Mock treatment was incubating cells at room temperature for 20 min. After treatment, cells were recovered for 60 min at 30℃. Protein was extracted by soaking in 0.1M NaOH followed by boiling in Laemmli buffer. Lysates were run on 4–20% SDS-PAGE stain-free gel, and imaged after UV activation to image total protein content. The gel was then transfered to nitrocellulose membrane. Dyed ladder was used as a guide to cut the membrane in half at approximately 50 kilodaltons (kDa). The lower part of the membrane was blotted for Hsp26 using a native antibody, a kind gift from Johannes Buchner. The upper half of the membrane was blotted for FLAG peptide with anti-FLAG

(Proteintech 66008–2-ig). Western blots were performed using the 1 hr Western Kit from GeneScript (L00204 and L00205).

## RNA-seq

### Sample preparation (ionophore)

(Data shown in *Figure 3—figure supplement 4*) Cells were grown as described in 'Growth and stress conditions' section above, resuspended in 1 mL of freshly prepared calibration curve buffer plus ionophore, and equilibrated for 15 min before being heat stressed at 42°C for 20 min. Cells were resuspended in SC media and allowed to recover for 5 min before being harvested, resuspended and flash frozen in lysis buffer (20 mM Tris pH 8, 140 mM KCl, 1.5 mM MgCl$_2$, 1% Triton-X100). Two biological replicates were collected and averaged.

### Sample preparation (media)

(Data shown in *Figure 3*, *Figure 3—figure supplement 1*, *Figure 3—figure supplement 2*, and *Figure 3—figure supplement 3*) Cells were grown as described in 'Growth and stress conditions' section above, resuspended in SC media with no pH adjustment (pH 4.0, acidification allowed), or adjusted to pH 7.5 using 2M KOH (acidification prevented). The following were then added to control translation state (all concentrations are final concentrations): 2% glucose (translation ongoing), 200 μg/mL cycloheximide (translation blocked), or nothing (0% glucose, translation blocked). Cells were heat stressed (42°) or mock-treated (room temperature) for 20 min, spun down at 3000 g for 1 min, and flash-frozen.

### Library preparation (ionophore)

Total cellular RNA was extracted using hot acid-phenol extraction and the resulting RNA was chemically fragmented. Samples were barcoded using a 3' adaptor with a unique sequence corresponding to each sample, and then pooled for downstream processing, as described in *Shishkin et al., 2015*. rRNA was depleted from the pooled samples using the Illumina Ribo-Zero Gold rRNA Removal Kit for Yeast (MRZY1306). Sequencing was performed at the Functional Genomics Core at the University of Chicago. Detailed protocol for library preparation is available; see *Shishkin et al., 2015*.

### Library preparation (media)

Total cellular RNA was extracted from cells using the Zymo Direct-Zol kit (catalog number R2051). RNA was additionally treated with Turbo DNase (Invitrogen, catalog number AM2238), and libraries were made from the resulting material using the Illumina TruSeq Library Prep Kit without poly(A) selection.

### Data processing

Processed data for the ionophore samples were generated from raw sequencing reads by identification with the unique sample bar code (allowing at most one mismatch) using custom scripts and then pseudo-aligned, without further processing, using kallisto (*Bray et al., 2016*) to an in-house generated S288C reference transcriptome including rRNA. The kallisto index was built with standard parameters, quantification was run with the command kallisto quant -i < index file> `--single` -b 100 -o < output file> -l 380 s 100 t 4 < data file>. Output per-gene normalized abundance estimates (transcripts per million, tpm) were used for all downstream analysis. Processed data for the media samples were generated from the raw reads directly with kallisto in paired-end mode with 100 iterations of the bootstrap algorithm.

### Heat shock genes

Genes upregulated during heat shock were curated by combining a list of Hsf1 targets from *Pincus et al., 2018* and Hsf1 targets and Msn2/4 targets from *Solís et al., 2016*.

### Stress transcription factor determination

Genes upregulated during stress were assigned to either Hsf1 or Msn2/4 as in *Solís et al., 2016*; *Pincus et al., 2018*. Briefly, the Msn2/4 genes were identified as genes that had a conserved Msn2/4

binding site in the upstream promoter and which were upregulated during heat stress in a strain of yeast where Hsf1 had been acutely deactivated. Hsf1 target genes were determined by differential expression after Hsf1 inactivation using a combination of transcript sequencing (RNA-seq), chromatin immunoprecipitation sequencing (ChIP-Seq), and native elongating transcript sequencing (NET-Seq). For *Figure 3D and E*, transcription factors were identified using the YeTFaSCo database (*de Boer and Hughes, 2012*) to generate a list of proteins that have annotated DNA binding motifs (259 genes); the regulon of each transcription factor were determined by using the YeastMine database to generate a list of genes which had previously been shown to be reguated by each gene. The database includes interactions determined both during heat shock and non-heat shock conditions; *Figure 3D* includes only transcription factors which had been assessed under heat shock conditions; *Figure 3E* includes the regulons of other know stress-associated transcription factors which were determined under non-heat shock conditions. For both figures only regulons with four or more genes were considered (minimum 11 genes, maximum 1844 genes, median 72 genes), and the genes under control of Hsf1 or Msn2/4 were excluded from other regulons.

## qPCR

Total cellular RNA was extracted from cells using the Zymo Direct-Zol kit (catalog number R2051). 100–200 ng of RNA were reverse-transcribed (iScript cDNA synthesis kit; catalog number 1708891) using gene-specific primers. The resulting DNA was then used as a template for qPCR (idt Prime-Time Gene Expression Master Mix; catalog number 1055770). For *SSA4*, primers and probes against mCherry were used to detect the transcript; for all other genes assayed the native sequence was detected. All transcript abundances are either expressed as a ratio to a control gene (*TUB2*) in the same sample relative to the same value in unstressed cells (*Figure 2E*), or as the ratio to a control gene (*TUB2*) in acidified to non-acidified cells (induction ratio, *Figure 3D*).

## Measuring fitness

### Relative growth rate

Competitive growth assays rely on tracking the relative size of two populations of cells as a function of time, and differences in growth rate are inferred from these data. The ratio of two populations, for example pHluorin-expressing (pH) and wild-type (wt) populations, as a function of time is given by the following equation:

$$\log\left(\frac{n_{pH}(t)}{n_{wt}(t)}\frac{n_{wt}(0)}{n_{pH}(0)}\right) = (r_{pH} - r_{wt}t) \tag{3}$$

where $n_x(t)$ is the number of cells of type x at time t, $r_x$ is the instantaneous growth rate (in units of $t^{-1}$), and $\frac{n_{pH}(0)}{n_{wt}(0)}$ is the initial mixing fraction. This equality is true assuming constant exponential growth, which indicates that our data are valid at least for the early stages of recovery; we only fit the linear portion of the data to ensure the validity of this assumption. For cells stressed without ionophore treatment, this was all timepoints less than 100 min, for cells stressed with ionophore this was all timepoints less than 160 min (this difference correlates roughly with the delay in induction we observe after ionophore treatment and is consistent across all pHs). See *Figure 6—figure supplement 1F* for all fits. We can use this equation to calculate the difference in growth rate, that is, the fitness loss, for each population of cells having experienced stress at a different intracellular pH. This value is expressed as a difference to arrested growth (maximal fitness loss) in *Figure 6B*.

The reference population (subscript *wt* in the above equation) is wild-type cells growing exponentially ('spike' or 'spike-in'), which are distinguishable from the pHluorin-expressing strains because they are are not significantly fluorescent in either pHluorin channel. Using a mixture of log-growing unlabeled and stressed labeled cells allows us to compare directly between the different pH and temperature combinations, as all the measured fitness loss values are relative to the same reference. It also implies that the difference $r_{pH} - r_{wt}$ will be either zero or negative, since the treatments being compared (pH manipulation either with or without heat shock) will in general only decrease the growth rate from maximal. To ensure that the pH manipulation itself was minimally stressful, the relative growth of pH-manipulated cells, which experienced 35 min at room temperature in calibration curve buffer with ionophore, was calculated and was found to be extremely close to 0 for all pH values considered (see *Figure 6—figure supplement 1D*, 'RT (mock)' row).

To control for possible additional, strain-specific differences, we also calculated the relative growth rate when both the wild-type and yCGT028 cells were treated identically ('mix' or 'mix-in'); this value was also found to be nearly zero in every condition examined (see *Figure 6—figure supplement 1D*, 'Mix-in' column).

### Determination of budded fraction

We first computationally isolated the labeled, stressed cells, and then for this population looked at the distribution of values in the Forward Scatter Width channel. It has been shown that values in this channel correspond most closely to cellular volume and size (*Tzur et al., 2011*; *Hoffman, 2009*) because the measurement represents the amount of time spent passing in front of the interrogating laser. We note that there are two populations of cells, which we assign to budded (larger) and unbudded (smaller) cells (*Figure 6—figure supplement 1A*, density plot). This approach has been previously used to discriminate budded and unbudded cells (*Hewitt and Nebe-Von-Caron, 2004*). Tracking the fraction of budded cells as a function of time gives information about cell cycle re-entry in a fashion analogous to the manual counting of budded and unbudded cells as previously performed (*Rowley et al., 1993*).

To verify this labeling, we sorted cells into two populations based on the forward scatter pulse width into 95% ethanol to fix, and then visualized the fixed cells using light microscopy; *Figure 6—figure supplement 1A* shows sorting parameters and representative microscopy images. Cells from both populations were blindly scored as either budded (containing an obvious bud that is at least 1/4 the size of the mother cell) or unbudded (having no bud). Full quantification is shown in *Figure 6—figure supplement 1B*. Fixed cells were then stained with Sytox to assess cell cycle position following a published protocol (*Rosebrock, 2017*), and DNA content was analyzed by fluorescence intensity using flow cytometry. The 'budded' population contained more cells in the 2x DNA peak, indicating that they were doubling their DNA and were thus actively growing; see *Figure 6—figure supplement 1C*.

### Code and data analysis

All data analysis was performed with R (*R Development Core Team, 2017*) using packages from the tidyverse (*Wickham, 2017*). Plots were made with ggplot2 (*Wickham, 2009*). Custom packages can be found on GitHub (*Triandafillou, 2020a*; https://github.com/ctriandafillou/flownalysis; copy archived at https://github.com/elifesciences-publications/flownalysis; *Triandafillou, 2020b*; https://github.com/ctriandafillou/cat.extras; copy archived at https://github.com/elifesciences-publications/cat.extras). Raw data and scripts processing it to produce all figures that appear in this work are available on Data Dryad: doi:10.5061/dryad.zgmsbcc6v.

In general, summary lines on plots are fits of the log-transformed data with the form:

$$\text{foldchange} = \frac{a}{1 + \exp(-b(\text{time} - c))} + d \tag{4}$$

where a, b, c, and d are fitting parameters, and d is constrained to be greater than or equal to 1. The exception to this are *Figure 1B*; *Figure 2A,C*; *Figure 6E*; and *Figure 5—figure supplement 1A*, which are moving averages.

### Statistical testing

Statistical significance was determined with either the Welch two-sample t-test (*Figure 2E*) or the Mann-Whitney U-test (Wilcoxon rank sum test) (*Figure 5G* and *Figure 3C*). All tests were performed using the stats package in the R programming language (*R Development Core Team, 2017*).

## Acknowledgements

Research reported in this publication was supported by the National Institute of Biomedical Imaging and Bioengineering of the National Institutes of Health (NIH) under Award Number T32EB009412, and the National Science Foundation Graduate Research Fellowship under Grant No. DGE-1144082. CDK was supported by the National Institute of General Medical Sciences (NIGMS) of the NIH, award number T32 GM007183. ARD acknowledges support from the NIGMS of the NIH award

numbers R01 GM109455 and R35 GM136381. DAD acknowledges support from the NIGMS of the NIH, award numbers R01 GM126547 and R01 GM127406, and from the US Army Research Office, award number W911NF-14-1-0411. The authors thank the University of Chicago Flow Cytometry Core for help with flow cytometry data collection and the Functional Genomics Core at the University of Chicago for assistance with sequencing. The authors also acknowledge members of the Drummond and Dinner labs for helpful comments and discussions.

## Additional information

### Funding

| Funder | Grant reference number | Author |
|---|---|---|
| National Institutes of Health | GM126547 | D. Allan Drummond |
| National Institutes of Health | GM127406 | D. Allan Drummond |
| Army Research Office | W911NF-14-1-0411 | D. Allan Drummond |
| National Institutes of Health | GM109455 | Aaron R Dinner |
| National Institutes of Health | T32EB009412 | Christopher D Katanski |
| National Institutes of Health | T32GM007183 | Catherine G Triandafillou |
| National Science Foundation | DGE-1144082 | Catherine G Triandafillou |
| National Institutes of Health | GM136381 | Aaron R Dinner |

The funders had no role in study design, data collection and interpretation, or the decision to submit the work for publication.

### Author contributions

Catherine G Triandafillou, Conceptualization, Data curation, Software, Formal analysis, Investigation, Visualization, Methodology, Writing - original draft, Writing - review and editing; Christopher D Katanski, Investigation; Aaron R Dinner, Supervision, Funding acquisition, Writing - original draft, Writing - review and editing; D Allan Drummond, Conceptualization, Resources, Supervision, Funding acquisition, Investigation, Methodology, Writing - original draft, Writing - review and editing

### Author ORCIDs

Catherine G Triandafillou https://orcid.org/0000-0002-6715-3795
Aaron R Dinner https://orcid.org/0000-0001-8328-6427
D Allan Drummond https://orcid.org/0000-0001-7018-7059

### Decision letter and Author response

Decision letter https://doi.org/10.7554/eLife.54880.sa1
Author response https://doi.org/10.7554/eLife.54880.sa2

## Additional files

### Supplementary files

• Transparent reporting form

### Data availability

Sequencing data have been deposited in GEO under accession codes GSE143292 and GSE152916. Raw and processed flow cytometry data, raw qPCR and translation data to reproduce all figures have been deposited to Dryad.

The following datasets were generated:

| Author(s) | Year | Dataset title | Dataset URL | Database and Identifier |
|---|---|---|---|---|
| Triandafillou CT, Katanski CD, Dinner AR, Drummond DA | 2020 | Transient intracellular acidification regulates the core transcriptional heat shock response | http://dx.doi.org/10.5061/dryad.zgmsbcc6v | Dryad Digital Repository, 10.5061/dryad.zgmsbcc6v |
| Triandafillou CT, Katanski CD, Dinner AR, Drummond DA | 2020 | Transient intracellular acidification regulates the core transcriptional heat shock response | https://www.ncbi.nlm.nih.gov/geo/query/acc.cgi?acc=GSE143292 | NCBI Gene Expression Omnibus, GSE143292 |
| Triandafillou CT, Katanski CD, Dinner AR, Drummond DA, Triandafillou CT, Katanski CD, Dinner AR, Drummond DA | 2020 | Transient intracellular acidification regulates the core transcriptional heat shock response | https://www.ncbi.nlm.nih.gov/geo/query/acc.cgi?acc=GSE152916 | NCBI Gene Expression Omnibus, GSE152916 |

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
