## [Decision Letter]

**Acceptance summary:**

We appreciate how your study furthers our understanding of the molecular signals that trigger a heat shock response in the model organism *Saccharomyces cerevisiae* and shows that a transient acidification of the cytoplasm can trigger a response even in the absence of translation. Acidification may contribute to phase separation of (existing, mature) proteins, which may in turn be a second trigger of the heat shock response. Because the heat shock response is a highly conserved and important cellular response, it is quite clear that the topic is of interest to the broad authorship of *eLife*.

**Decision letter after peer review:**

Thank you for submitting your article "Transient intracellular acidification regulates the core transcriptional heat shock response" for consideration by *eLife*. Your article has been reviewed by three peer reviewers, including Kevin J Verstrepen as the Reviewing Editor and Reviewer #1, and the evaluation has been overseen by Kevin Struhl as the Senior Editor. The following individual involved in review of your submission has agreed to reveal their identity: Kevin Morano (Reviewer #2).

The reviewers have discussed the reviews with one another and the Reviewing Editor has drafted this decision to help you prepare a revised submission.

We appreciate that your study challenges the current model for heat shock activation, showing that blocking translation either through carbon deprivation or through chemical inhibition still allows cells to mount a response, but only if they undergo a transient acidification of the cytoplasm.

As you can see in the individual reviews below, we are impressed with the overall quality of this study and excited about the possible implications. However, we also believe that there still are a few loopholes that warrant further exploration. In particular, we are not convinced that the current set of experiments establishes intracellular acidification as a direct trigger of the heat shock response, and we worry that alternative hypotheses have not been ruled out or at least acknowledged. There is a strong consensus among the reviewers that performing a transcriptome (RNAseq) analysis of cells progressing through a HSR in acidic or neutral pH, both while transcription is blocked or not, would help understand whether the effect of intracellular acidification is completely similar to the normal Hsf-driven HSR response, or instead is a somewhat different, more general response triggered by the low intracellular pH (eg as a consequence of general issues with transcription). We therefore ask you to perform this experiment and to critically discuss the results in the main text (not in the supplements).

In addition, we also ask to provide a more critical discussion of the results, acknowledging unanswered questions and possible alternative hypotheses.

Lastly, we suggest to also have a look at the individual reviews below. Whereas we do not think that it is necessary to address all issues listed, we do think that several of these other points may help to further improve the manuscript.

Reviewer #1:

In this study, Triandafillou and colleagues investigate the molecular signals that trigger a heat shock response in the model organism *Saccharomyces cerevisiae*. The current model for heat shock activation predicts that temperature-induced misfolding of newly synthesized proteins triggers the response, but the authors find that blocking translation either through carbon deprivation or through chemical inhibition still allows cells to mount a response, but only if they undergo a transient acidification of the cytoplasm. This acidification is shown to depend on an acidic environment, possibly because extracellular protons are able to diffuse through the membrane as it becomes unstable at higher temperatures. Acidification may contribute to phase separation of (existing, mature) proteins, which may in turn be a second trigger of the heat shock response.

Overall, this is an extremely interesting and thought-provoking study, summarized in an eloquently written, albeit somewhat wordy manuscript. Because the heat shock response is a highly conserved and important cellular response, it is quite clear that the topic is of interest to the broad authorship of *eLife*.

Although the scientific work is strong and solid, the authors could in some instances perhaps be a bit more critical and cautious with the interpretation of their results. While the hypothesis seems plausible, it would be good to also point out alternative explanations and pitfalls, and to clearly separate correlation from causation. A few extra sentences in the Results and Discussion may suffice.

For example, whereas glucose deprivation may indeed lead to translation arrest, it would be best to show this directly and to also consider the other effects, including a possible sudden drop in intracellular ATP and activation of various signaling pathways (main glucose repression pathway, cAMP pathway…), which could also contribute to the activation of the heat shock response. In that respect, I think that the experiments with cycloheximide provide a stronger basis, even though it is a much stronger perturbation of the cell, which in itself also increases the risks of undesirable side effects.

Second, the fact that a proper heat shock response correlates with transient acidification is interesting, but I am not convinced that "it dictates chaperone production" (seems to imply causation).

Third, while an attractive hypothesis, I am not sure whether there is enough proof to state that the acidification-dependent response indeed depends on phase separation of existing, mature proteins. Since acidification not only occurs during a heat shock, it might in fact be a more general trigger of a more general stress response, and while the phase-separation-theory looks plausible, one can imagine other sensors and pathways. In that sense, it would be interesting (but perhaps not necessary in this study) to carefully compare the cellular response (transcriptome) to acidification (eg using the ionophore) to that of a "normal" heat shock; and that of a heat shock in acidic medium to one in neutral medium. In the current manuscript, it would perhaps be best to explicitly state the that the phase-separation of mature Hsp70 clients is at this stage only a theory, and that other explanations should not yet be ruled out?

Reviewer #2:

The heat shock response (HSR) is an ancient and strongly conserved cellular program required for survival and adaptation to conditions leading to proteotoxic stress. While work in multiple experimental systems has established core tenets of the HSR, namely control of gene expression by the transcription factor Hsf1 that is induced/derepressed by protein misfolding and in some cases other cytotoxic stresses, a flurry of recent research activity in budding yeast has revealed more precise insights. Hsp70 is now recognized to be the primary, if not sole regulator of the Hsf1 activation switch, and nearly any condition that occupies or impairs Hsp70 can elicit the HSR. Additionally, the existence of temperature- and pH-specific protein dynamics in the yeast cytoplasm that impact Hsp70 and other chaperones raise the possibility of multiple sensing mechanisms that may channel into HSR activation. This manuscript by Triandafillou and colleagues challenges long-held assumptions that protein misfolding is required for HSR activation by heat shock and finds that it is not – transient intracellular acidification is required for the HSR and only when translation AND acidification are blocked does chaperone gene expression fail. Subsequent experiments additionally show that return to "normal" pH after HS is also required for maximal induction, and that these processes are fitness-promoting and therefore evolutionarily beneficial. Finally, transcriptome experiments are utilized to demonstrate that the observed effects are specific to the HSR.

Overall, the work reveals a previously unknown aspect of HSR regulation and is therefore novel and impactful. Moreover, the tools generated to carry out the study as well as the sophistication of the experiments (for example, single-cell correlation of HSR activation and intracellular pH) are particular strengths. Below I outline a few considerations and necessary clarifications I feel the authors need to address:

1) Figure 1B – the solid line sliding-window average appears to suggest that the pH is dropping prior to the application of heat shock. Is this just an "artifact" of the averaging process? Or perhaps lack of resolution between the addition of warm medium and immediate next reads in the flow cytometer?

2) With regard to physiological acidification, there is a great deal of investigation and discussion as to why this happens, but little about how it happens. While the ionophore- and medium-based manipulations are convincing, was thought given to using a complementary genetic approach? Mutants in proton antiporters, or even the PMA1 membrane ATPase can influence cytoplasmic pH. Identifying a mechanism to explain this important physiological effect seems germane, if not proximal, to the operational model the authors are trying to build.

3) Figure 2B – the authors immediately interpret the differential effects of glucose starvation as translational arrest. This is problematic for two reasons. Firstly, they do not go on to demonstrate translational arrest under these conditions until Figure 2D. Secondly, a great deal of physiological change is being experienced by cells experiencing glucose withdrawal, including translational arrest. I suggest that the authors flip the order of 2B and 2D and/or allow for several possible rationales for the failure to induce Ssa4 that are then narrowed down to translation via the experiments in 2D and 2E. It should also be pointed out that glucose withdrawal is known to decrease cytoplasmic pH via proton pump inhibition (Martínez-Muñoz, G. A., and Kane, P. (2008) Vacuolar and plasma membrane proton pumps collaborate to achieve cytosolic pH homeostasis in yeast. J. Biol. Chem. 283, 20309-20319), possibly confounding the use of glucose withdrawal solely to block translation.

4) The experiments shown in Figures 3D, 3F and 4B appear to suggest that ionophore treatment is not required to observe external pH-mediated effects on Ssa4 induction. Am I interpreting this correctly? If so, are the ionophore experiments necessary for the recovery questions?

5) While it is not apparent from the plot shown in Figure 6C, 6D appears to show a significant, though smaller in magnitude, induction ratio for the two Msn2 targets tested. The authors conclude that the pH effects they have observed are specific to Hsf1, but I think that's not so clear. Hsf1 is definitely more dependent than Msn2/4, but it is not exclusively dependent. Have the authors excluded the possibility that general transcriptional induction, i.e., Mediator-controlled rapid pol II recruitment, is not more widely affected? Showing the GAL genes are equally well induced at the same pHs would be an easy way to dispel this notion.

Reviewer #3:

This manuscript explores the functional consequences of the long noted observation that many different types of cells experience transient cytosolic acidification after heat shock. Specifically the authors use budding yeast to examine if cytosolic acidification is important for mounting the core transcriptional heat shock response mediated by Hsf1 and whether this phenomenon has any cell adaptive role during stress.

To do this, they develop a FACS method to simultaneously monitor the cytosolic pH and the induction of a reporter of the heat shock response (Ssa4-mCherry), over time, in heat shocked cell populations.

By blocking cytosolic acidification during heat shock by transferring cells to pH7.5 media they show that the heat shock response is blunted when translation is ongoing and abrogated when translation is blocked. This result is recapitulated using nigericin to manipulate cytosolic pH during heat shock.

The authors find that preventing cytosolic acidification during heat shock slows the return of cytosolic pH to a resting state during recovery and propose that delay is influencing the dynamics of chaperone production.

Finally, they show data that suggests that cytosolic acidification during heat shock promotes improved fitness of the cell population and that the lack of cytosolic acidification during heat shock affects Hsf1 regulated targets specifically.

The data in this manuscript seem to be of high quality, the manuscript is well written and is potentially of broad interest to readers of *eLife*. The speculation about the need for interplay between the heat shock transcriptional program and cytosolic acidification to promote yeast cell fitness during their dispersion by passerine birds is particularly enjoyable to consider.

A major concern however, is that this manuscript does not provide clear evidence that preventing cytosolic acidification during heat shock is directly influencing the Hsf1-mediated heat shock transcriptional program versus indirect effects of the temporal interplay between the pH stress responsive pathway driven by Rim101 (or a variety of other stress responsive pathways likely induced when cells are placed in a buffer containing 50 mM NaCl, 50 mM KCl, 50 mM MES, 50 mM HEPES, 100 mM ammonium acetate and 10 mM 2-deoxyglucose). Testing this direct causality would require finding a way of preventing cytosolic acidification independently of manipulating extracellular pH (Rim101 pathway is driven by external cues). Unfortunately, such a separation is non-trivial (there are many genes involved in pH homeostasis) and I am not aware of a simple way to separate the internal and external pH effect seen here. This may require a genetic screen to identify candidates to accomplish such an "unlinking". Another potential way to address this concern is to use RNAseq to identify the transcriptional changes associated with the experimental procedure and then test the dynamics of Ssa4 induction in cells unable to mount those changes.

Given the experimental procedure the authors employ is critical to make the conclusions drawn in this manuscript, they must rule out the influence of other (pH?) stress responsive transcriptional programs on the dynamics of the heat shock response, before this manuscript should be accepted in *eLife*.

---

## [Author Response]

We appreciate that your study challenges the current model for heat shock activation, showing that blocking translation either through carbon deprivation or through chemical inhibition still allows cells to mount a response, but only if they undergo a transient acidification of the cytoplasm.As you can see in the individual reviews below, we are impressed with the overall quality of this study and excited about the possible implications. However, we also believe that there still are a few loopholes that warrant further exploration. In particular, we are not convinced that the current set of experiments establishes intracellular acidification as a direct trigger of the heat shock response, and we worry that alternative hypotheses have not been ruled out or at least acknowledged.

We appreciate the support and appropriate skepticism. At no point have we claimed that acidification is a direct trigger; indeed, in the Discussion we hypothesize the opposite (that Hsf1 activation’s pH-dependence is indirect, through the substrates of Hsf1’s repressor, Hsp70). In revision, we have attended closely to clarifying the nature (hypothesis, speculation, synthesis) of the interpretations for the phenomena we report, and to affirming that the mechanism remains open. We hope the reviewers understand that we have articulated what we feel are the most plausible hypotheses, along with testable predictions.

For example, in the Discussion, we state: “Together, these studies indicate the existence of multiple proteins that undergo heat-triggered, pH-dependent condensation processes, producing assemblies which conditionally recruit Hsp70. Based on these observations, we make three hypotheses to be tested in future work. First, we hypothesize that pH- and temperature-dependent condensing proteins are the cause of the Hsf1 pH-dependence we observe when translation is silenced. Second, we predict that if such proteins are found, they will activate Hsf1 by recruiting Hsp70 in the condensed state. Third, we hypothesize that many such proteins exist, such that it should be possible to activate Hsf1 by triggering condensation of a single protein, but suppression of any single protein's condensation will not suppress Hsf1 activation.”

We end the following section with: “We underscore that these are predictions based on a synthesis of existing knowledge, and that the mechanistic basis for the pH sensitivity we report is now a crucial open question. It remains possible, for example, that Hsf1 directly senses pH and translational activity. A conceptual advantage of our model is that, unlike this example, it is built from empirically extant pieces.”

There is a strong consensus among the reviewers that performing a transcriptome (RNAseq) analysis of cells progressing through a HSR in acidic or neutral pH, both while transcription is blocked or not, would help understand whether the effect of intracellular acidification is completely similar to the normal Hsf-driven HSR response, or instead is a somewhat different, more general response triggered by the low intracellular pH (eg as a consequence of general issues with transcription). We therefore ask you to perform this experiment and to critically discuss the results in the main text (not in the supplements).

We have carried out a large set of new RNA-seq experiments to address these and related questions. These results are now central to the manuscript. Briefly, as described in our revised Materials and methods section, we performed RNA-seq on cells in media (as opposed to ionophore treated, as in the previous version) where acidification was either blocked or allowed, as shown in Figure 2. In order to determine the effect of translation state, we examined the transcriptional response to heat stress, in cells that acidified vs. those that didn’t, when translation was ongoing, blocked by cycloheximide treatment, or blocked by acute glucose withdrawal. We performed equivalent mock treatments (the same media and translation conditions without heat shock), and did every condition in biological duplicate (correlation between replicates was quite good, and are quantified in Figure 3—figure supplement 1A). In total, this amounted to 22 samples, with data now deposited in GEO under accession number GSE152916.

The results confirm our previous low-throughput findings. The major addition is that the pH sensitivity of the Hsf1 regulon is now remarkably clear, and clearly different than other regulons including Msn2/4.

In addition, we also ask to provide a more critical discussion of the results, acknowledging unanswered questions and possible alternative hypotheses.

As noted above, we have rewritten much of the Discussion to more clearly state hypotheses and distinguish them from our empirical results and from prior work.

Reviewer #1:In this study, Triandafillou and colleagues investigate the molecular signals that trigger a heat shock response in the model organism *Saccharomyces cerevisiae*. The current model for heat shock activation predicts that temperature-induced misfolding of newly synthesized proteins triggers the response, but the authors find that blocking translation either through carbon deprivation or through chemical inhibition still allows cells to mount a response, but only if they undergo a transient acidification of the cytoplasm. This acidification is shown to depend on an acidic environment, possibly because extracellular protons are able to diffuse through the membrane as it becomes unstable at higher temperatures. Acidification may contribute to phase separation of (existing, mature) proteins, which may in turn be a second trigger of the heat shock response.Overall, this is an extremely interesting and thought-provoking study, summarized in an eloquently written, albeit somewhat wordy manuscript. Because the heat shock response is a highly conserved and important cellular response, it is quite clear that the topic is of interest to the broad authorship of eLife.Although the scientific work is strong and solid, the authors could in some instances perhaps be a bit more critical and cautious with the interpretation of their results. While the hypothesis seems plausible, it would be good to also point out alternative explanations and pitfalls, and to clearly separate correlation from causation. A few extra sentences in the Results and Discussion may suffice.

We appreciate these helpful comments. In the Discussion, we have tried to be even clearer when advancing hypotheses versus merely providing an interpretation of results. The reviewer’s important comment about separating correlation from causation is echoed in another comment below where we respond more fully.

For example, whereas glucose deprivation may indeed lead to translation arrest, it would be best to show this directly and to also consider the other effects, including a possible sudden drop in intracellular ATP and activation of various signaling pathways (main glucose repression pathway, cAMP pathway…), which could also contribute to the activation of the heat shock response. In that respect, I think that the experiments with cycloheximide provide a stronger basis, even though it is a much stronger perturbation of the cell, which in itself also increases the risks of undesirable side effects.

We have measured (using incorporation of radiolabeled amino acids) translation during glucose withdrawal, under conditions both where the cytoplasm can acidify and where it can’t; these data are in Figure 2B. They show that translation is halted when cells are deprived of glucose (as known), and that this effect is independent of intracellular acidification.

The reviewer’s point that there are other effects of glucose withdrawal is certainly relevant. To this end we examined both glucose withdrawal and cycloheximide treatment in the sequencing experiment we performed for this revision (new Figure 3). We were able to correct for the general effect of glucose withdrawal (which are quantitatively similar to those observed previously, see a direct comparison in Figure 3—figure supplement 3B) by normalizing the heat-shocked abundances to mock-treated abundances (see Figure 3—figure supplement 3A). This demonstrates that there is an additional effect on Hsf1 genes by acidification that goes beyond the effects produced by glucose withdrawal.

Second, the fact that a proper heat shock response correlates with transient acidification is interesting, but I am not convinced that "it dictates chaperone production" (seems to imply causation).

We agree. The causal effect of the return to resting pH is clear at the population level where we carried out a perturbation experiment, but has not been demonstrated at the single-cell level where this phrase appears. We have revised the text to say (with context): “Return to the resting pH predicts the dynamics of chaperone production. Acidification, either simultaneous with or following heat stress, followed by return to the resting pH appears to be required for robust induction of chaperones after heat stress.”

Third, while an attractive hypothesis, I am not sure whether there is enough proof to state that the acidification-dependent response indeed depends on phase separation of existing, mature proteins. Since acidification not only occurs during a heat shock, it might in fact be a more general trigger of a more general stress response, and while the phase-separation-theory looks plausible, one can imagine other sensors and pathways.

In revision, we have redoubled our efforts to ensure that no confusion arises between a result and a hypothesis. We hypothesize that biomolecular condensation is involved; at no point do we state that this has been shown or that all other hypotheses have been ruled out. We now explicitly state, “the mechanistic basis for the pH sensitivity we report is now a crucial open question.”

In that sense, it would be interesting (but perhaps not necessary in this study) to carefully compare the cellular response (transcriptome) to acidification (eg using the ionophore) to that of a "normal" heat shock; and that of a heat shock in acidic medium to one in neutral medium.

We have carried out extensive new experiments using RNA-seq which answer some of these questions. We compare the heat shock response in both acidic and basic media, and find that under conditions of high translation there is no detectable difference in the induction of heat shock proteins (Figure 3C). However, we find that under conditions where translation has been blocked, that the induction of heat shock proteins is significantly lower in cells that cannot acidify, which we interpret as a failure to activate Hsf1 under conditions where cells are both non-translating and cannot acidify.

In these experiments we did not use an ionophore and so cannot (and do not) make any claims about the broader response to acidification in the absence of other perturbations.

In the current manuscript, it would perhaps be best to explicitly state the that the phase-separation of mature Hsp70 clients is at this stage only a theory, and that other explanations should not yet be ruled out?

In the Discussion, we now state explicitly: “We underscore that these are predictions based on a synthesis of existing knowledge, and that the mechanistic basis for the pH sensitivity we report is now a crucial open question.”

Reviewer #2:[…]Overall, the work reveals a previously unknown aspect of HSR regulation and is therefore novel and impactful. Moreover, the tools generated to carry out the study as well as the sophistication of the experiments (for example, single-cell correlation of HSR activation and intracellular pH) are particular strengths. Below I outline a few considerations and necessary clarifications I feel the authors need to address:1) Figure 1B – the solid line sliding-window average appears to suggest that the pH is dropping prior to the application of heat shock. Is this just an "artifact" of the averaging process? Or perhaps lack of resolution between the addition of warm medium and immediate next reads in the flow cytometer?

We applaud the eagle eye of the reviewer! The effect referred to is a combination of the smoothing, and the fact that the increase in temperature is performed manually and varies slightly between experiments. In some cases, the application of increased temperature by addition of warm medium was performed slightly earlier, leading to a seeming early drop in pH when plotted on a common axis. We have added an explanation of this effect in the text to avoid confusion.

2) With regard to physiological acidification, there is a great deal of investigation and discussion as to why this happens, but little about how it happens. While the ionophore- and medium-based manipulations are convincing, was thought given to using a complementary genetic approach? Mutants in proton antiporters, or even the PMA1 membrane ATPase can influence cytoplasmic pH. Identifying a mechanism to explain this important physiological effect seems germane, if not proximal, to the operational model the authors are trying to build.

We agree that understanding the mechanism by which acidification occurs is interesting and relevant to the topic, we feel that it is outside the scope of this work. This issue has been previously explored in the literature (for example in Coote, Cole and Jones, 1994) and we’ve included a slightly more extended discussion of these results.

3) Figure 2B – the authors immediately interpret the differential effects of glucose starvation as translational arrest. This is problematic for two reasons. Firstly, they do not go on to demonstrate translational arrest under these conditions until Figure 2D.

We understand the reviewer’s point and our revision is substantially clearer. Translational arrest due to glucose withdrawal is well-established per citations in the paper, and we do not claim that glucose withdrawal only causes translational arrest. In revision we write: “To test whether acidification still promoted the stress response even under conditions where the concentration of newly synthesized polypeptides would be sharply limited, we first used brief glucose withdrawal, a physiologically relevant condition which is known to rapidly and reversibly inhibit translation of most cellular mRNAs (Ashe, 2000)” We also now confirm this arrest under our precise conditions in Figure 2B.

Secondly, a great deal of physiological change is being experienced by cells experiencing glucose withdrawal, including translational arrest. I suggest that the authors flip the order of 2B and 2D and/or allow for several possible rationales for the failure to induce Ssa4 that are then narrowed down to translation via the experiments in 2D and 2E.

We agree that glucose withdrawal is not a specific perturbation of translation, and use cycloheximide both here and in the new sequencing experiments in Figure 3, where the differences between glucose withdrawal and cycloheximide treatment become clearer. In introducing the cycloheximide experiments in Figure 2E, we now write “To directly inhibit translation without nutrient withdrawal…”

It should also be pointed out that glucose withdrawal is known to decrease cytoplasmic pH via proton pump inhibition (Martínez-Muñoz, G. A., and Kane, P. (2008) Vacuolar and plasma membrane proton pumps collaborate to achieve cytosolic pH homeostasis in yeast. J. Biol. Chem. 283, 20309-20319), possibly confounding the use of glucose withdrawal solely to block translation.

These are important details. We investigate the drop in pH induced by glucose withdrawal on the short timescales that we employ it in Figure 2A and C. Brief (in this case ~10 minutes) pre-treatment in 0% glucose SC (pH 4) is not enough to cause acidification on its own, and cells acidify normally when heat shocked as shown in Figure 2C. We are of course aware that glucose withdrawal cannot be used to solely block translation; we use cycloheximide as a more targeted inhibitor (Figure 2E and Figure 3).

4) The experiments shown in Figures 3D, 3F and 4B appear to suggest that ionophore treatment is not required to observe external pH-mediated effects on Ssa4 induction. Am I interpreting this correctly? If so, are the ionophore experiments necessary for the recovery questions?

You are correct—ionophore is not required to merely prevent or allow acidification. However, we were interested not only in the binary distinction between acidifying and not, but also the underlying quantitative question, that is, what is the quantitative relationship between acidification and the response? Is there a certain degree of acidification or a particular pH that was required for a response? In order to address this question (and subsequent questions about recovery), we performed the experiments with the ionophore. We were also interested in the important question of whether acidification without concomitant heat shock had an effect on cells, either in terms of heat shock protein production or cellular fitness. In order to create acidification without heat shock, we found it necessary to use ionophore treatment.

5) While it is not apparent from the plot shown in Figure 6C, 6D appears to show a significant, though smaller in magnitude, induction ratio for the two Msn2 targets tested. The authors conclude that the pH effects they have observed are specific to Hsf1, but I think that's not so clear. Hsf1 is definitely more dependent than Msn2/4, but it is not exclusively dependent. Have the authors excluded the possibility that general transcriptional induction, i.e., Mediator-controlled rapid pol II recruitment, is not more widely affected? Showing the GAL genes are equally well induced at the same pHs would be an easy way to dispel this notion.

We agree that the question of whether there is a more general effect (or even more of an effect on Msn2/4 genes) is important, and thus we considered this in the design of the sequencing experiment we did for the revision. We heat stressed cells with and without acidification (using media conditions rather than ionophore to mitigate concerns about the ionophore affecting the cells in unintended ways); and with translation ongoing, blocked by cycloheximide, or blocked by glucose withdrawal. The data are summarized in Figure 3 and accompanying supplemental figures.

As we now note in the text, “Because the Msn2/4 regulon continues to induce robustly independent of pH and translational status, a broader effect of pH on transcriptional processes cannot explain Hsf1's sensitivity.” The new Figure 3A (bottom right) demonstrates that a subset of the heat shock response continues to be induced to high levels in the presence of translational inhibitors at pH 7.5, and Figure 3C shows that the Msn2/4 regulon is virtually unaffected by lack of acidification. This indicates an effect that is, at least between Msn2/4 and Hsf1, specific to Hsf1.

To address whether this effect was truly specific to Hsf1, or instead affected other transcription factors, we took two separate approaches. The first was to compare the per-gene sensitivity to acidification for Hsf1 genes, Msn2/4 genes, and all other genes that were upregulated in our dataset in response to heat shock. This is shown in Figure 3—figure supplement 2C. When translation is arrested, only the Hsf1 genes are preferentially induced by acidification. Under conditions of glucose withdrawal there is a slight increase in pH-sensitivity for Msn2/4 and “other” genes, and a marked sensitivity of Hsf1 genes—thus there may be some more general effect of acidification when cells experience glucose withdrawal; this is in line with previous work showing that acidification is part of the response to acute energy depletion (for example in Munder et al., 2016 and Joyner et al., 2016).

The second approach we took was to look for pH-sensitivity in the regulons of all annotated transcription factors, and here Hsf1 was the only transcription factor that showed sensitivity to acidification during heat shock (Figure 3D).

We identified two other transcription factors which showed behavior opposite Hsf1 (Ifh1 and Ste12): acidification-dependent repression (rather than activation) when translation was blocked. To our knowledge this has not been reported before.

Reviewer #3:[…]A major concern however, is that this manuscript does not provide clear evidence that preventing cytosolic acidification during heat shock is directly influencing the Hsf1-mediated heat shock transcriptional program versus indirect effects of the temporal interplay between the pH stress responsive pathway driven by Rim101 (or a variety of other stress responsive pathways likely induced when cells are placed in a buffer containing 50 mM NaCl, 50 mM KCl, 50 mM MES, 50 mM HEPES, 100 mM ammonium acetate and 10 mM 2-deoxyglucose). Testing this direct causality would require finding a way of preventing cytosolic acidification independently of manipulating extracellular pH (Rim101 pathway is driven by external cues). Unfortunately, such a separation is non-trivial (there are many genes involved in pH homeostasis) and I am not aware of a simple way to separate the internal and external pH effect seen here. This may require a genetic screen to identify candidates to accomplish such an "unlinking". Another potential way to address this concern is to use RNAseq to identify the transcriptional changes associated with the experimental procedure and then test the dynamics of Ssa4 induction in cells unable to mount those changes.Given the experimental procedure the authors employ is critical to make the conclusions drawn in this manuscript, they must rule out the influence of other (pH?) stress responsive transcriptional programs on the dynamics of the heat shock response, before this manuscript should be accepted in eLife.

We agree with the reviewer that the specificity of the effect to Hsf1 and potential effects on other pathways needed to be more clearly demonstrated. To that end, and consistent with the reviewer’s specific suggestion along with the summary feedback, we performed RNAseq to look at the global effect of acidification during heat shock, both when translation was ongoing, or when blocked by either treatment with cycloheximide or withdrawal of glucose.

We found that when translation was blocked by either means, acidification specifically promoted the induction of the Hsf1-exclusive regulon. This is evident in the CDF in Figure 3C. However, we wanted to more fully and directly address whether there was a global effect of acidification. To address whether the effect we observed (promotion by acidification) was truly specific to Hsf1 or was instead more general, we took two approaches. The first was to compare the per-gene sensitivity to acidification for Hsf1 genes, Msn2/4 genes, and all other genes that were upregulated in our dataset in response to heat shock. This is shown in Figure 3—figure supplement 2C. When translation is arrested, only the Hsf1 genes are preferentially induced by acidification. Under conditions of glucose withdrawal there is a slight increase in pH-sensitivity for Msn2/4 and “other” genes, and a marked sensitivity of Hsf1 genes — thus there may be some more general effect of acidification when cells experience glucose withdrawal; this is in line with previous work showing that acidification is part of the response to acute energy depletion (for example in Munder et al., 2016 and Joyner et al., 2016).

The second approach we took was to look for pH-sensitivity in the regulons of all annotated transcription factors, and here Hsf1 was the only transcription factor that showed sensitivity to acidification during heat shock (Figure 3D). The effect of acidification on the Rim101 regulon during heat shock (as mentioned by the reviewer) is shown in Figure 3—figure supplement 3C; it looks similar to all genes, and does not seem to be affected in a pH-specific manner.